# Infectious polymorphic toxins delivered by outer membrane exchange discriminate kin in myxobacteria

Christopher N Vassallo[1], Pengbo Cao[1], Austin Conklin[1], Hayley Finkelstein[2], Christopher S Hayes[2], Daniel Wall[1]*

[1]Department of Molecular Biology, University of Wyoming, Laramie, United States; [2]Department of Molecular, Cellular and Developmental Biology, University of California, Santa Barbara, United States

**Abstract** Myxobacteria are known for complex social behaviors including outer membrane exchange (OME), in which cells exchange large amounts of outer membrane lipids and proteins upon contact. The TraA cell surface receptor selects OME partners based on a variable domain. However, *traA* polymorphism alone is not sufficient to precisely discriminate kin. Here, we report a novel family of OME-delivered toxins that promote kin discrimination of OME partners. These SitA lipoprotein toxins are polymorphic and widespread in myxobacteria. Each *sitA* is associated with a cognate *sitI* immunity gene, and in some cases a *sitB* accessory gene. Remarkably, we show that SitA is transferred serially between target cells, allowing the toxins to move cell-to-cell like an infectious agent. Consequently, SitA toxins define strong identity barriers between strains and likely contribute to population structure, maintenance of cooperation, and strain diversification. Moreover, these results highlight the diversity of systems evolved to deliver toxins between bacteria.

*For correspondence: dwall2@ uwyo.edu

Competing interests: The authors declare that no competing interests exist.

## Introduction

Cooperative, social organisms benefit by resource sharing and division of labor between individuals in a population. These behaviors entail directing beneficial action toward kin, often at a fitness cost to the actor. Relatedness between individuals must be high for such cooperative action to remain evolutionarily viable (*Hamilton, 1964a*, *1964b*). This requires that social organisms recognize their kin, and direct preferential action toward them (kin discrimination). The mechanisms by which social microbes recognize and direct benefits toward kin cells are not well understood. However, insights in this area will help us to understand the organization of microbes into social groups and the behaviors that maintain cooperation despite seemingly opposing evolutionary pressures to be selfish.

The soil bacterium *Myxococcus xanthus* is a model organism for the study of social behavior and cooperation (*Cao et al., 2015*). Myxobacterial populations divide labor and share resources during coordinated behaviors such as swarming, predation, and starvation-induced fruiting body development. Their social lifestyle, which includes multicellular development by an aggregation strategy, requires that they direct cooperative behavior towards their clonemates or close kin. One such cooperative behavior is outer membrane exchange (OME). During OME, swarming cells in a population simultaneously donate and receive prodigious amounts of outer membrane (OM) material between one another during cell contact. Exchanged material includes membrane lipids, lipoproteins and lipopolysaccharide (*Nudleman et al., 2005*; *Wei et al., 2011*; *Vassallo et al., 2015*; *Pathak et al., 2012*). The mechanism for exchange is thought to involve transient OM fusion catalyzed by the OM receptor TraA and an associated protein, TraB (*Cao et al., 2015*; *Pathak et al., 2012*). Our model

**eLife digest** Most people do not think of bacteria as having a social life. However, some groups, such as myxobacteria, are highly cooperative. Although these microbes exist as individual cells, they can also move and hunt in coordinated packs and when nutrients are low, about a million cells come together to build spore-filled structures. To do so, myxobacteria need to recognize their sibling cells among the vast number of different species of microbes found in soil.

One way that the bacteria recognise their kin is by displaying a variable cell surface protein, called TraA, that identifies other individuals that display the same protein on their surface. Upon recognition, cells exchange resources by briefly fusing their outer membranes. This allows bacteria to help to rejuvenate damaged sibling cells by delivering healthy cell components to them. Now, using a genetic approach, Vassallo et al. present evidence that bacteria can also exchange toxins.

The newly identified toxin-exchange system works alongside the TraA kin recognition system to allow myxobacteria to recognize and verify their true sibling cells in diverse environments. The cells involved in the exchange must contain matching immunity proteins to survive the interaction – thus the exchange does not harm sibling cells. Strikingly, once the toxic proteins are delivered, they can be passed on to other cells by a series of transfers, much like an infection spreads throughout a population.

The study performed by Vassallo et al. provides a new framework for understanding how microbes recognize their kin to build a community. These insights will help investigators to explore other microbial ecosystems, including those found inside the human body. Additionally, the results also suggest ways in which cells can be engineered to specifically recognize other cells to transfer materials between them. This system could be adapted to program different cell types so that they interact with specific partners and perform complex tasks.

predicts that transient OM fusion between two cells enables the lateral diffusion of OM lipids and proteins between OMs until cells move apart and the membranes are again separated (*Cao et al., 2015*). This process occurs constitutively on surfaces and facilitates efficient OM homogenization of populations with heterogeneous OMs (*Wei et al., 2011*). Exchange of fluorescent OM lipoprotein reporters, as well as endogenous OM lipoproteins, demonstrates that nearly all recipient cells receive substantial amounts of cargo protein within two hours of co-culture (*Nudleman et al., 2005*; *Wei et al., 2011*). Furthermore, cells with lethal defects in lipopolysaccharide biosynthesis can be sustained in a population by OME with wild-type (WT) donors (*Vassallo et al., 2015*). Based on this observation, OME is hypothesized to help physiologically heterogeneous populations move toward homeostasis and buffer cell damage to support synchronized and cohesive group behaviors (*Vassallo et al., 2015*; *Vassallo and Wall, 2016*). This robust system for sharing cellular goods must be discriminately targeted to closely related cells – that is, clonemates. Otherwise, this organism risks donating private goods to competing, non-kin genotypes. In this regard, we previously showed that TraA has a variable domain that specifies recognition between cells through homotypic interactions (*Pathak et al., 2013*; *Cao and Wall, 2017*). Thus, myxobacteria with divergent, incompatible TraA receptors do not engage in OME. *traA* is therefore a greenbeard gene in that it allows myxobacteria to identify cells with identical alleles and to direct beneficial treatment toward those cells (*Dawkins, 1976*). Greenbeard alleles do not exclusively recognize kin genotypes, but instead recognize any genotype that possesses the same allele (kind discrimination) (*Queller, 2011*; *Strassmann et al., 2011*). Indeed, although TraA sequence diversity in the variable domain is high, some non-kin genotypes share compatible *traA* alleles (*Pathak et al., 2013*). In fact, some *Myxococcus* isolates that antagonize one another in co-culture possess the same *traA* alleles (*Pathak et al., 2013*). Based on this observation, we hypothesized that there are additional genetic determinants that more precisely discriminate kin during social interactions.

Bacterial kin discrimination is often mediated by antagonism toward non-kin. One mechanism that bacteria use to this end is the delivery of polymorphic toxins between cells in close contact (*Zhang et al., 2012*; *Ruhe et al., 2013*; *Cardarelli et al., 2015*; *Wenren et al., 2013*). These toxins usually share homology in species-specific amino-terminal domains required for presentation and/or

delivery, but vary in their carboxy-terminal toxin domains (*Zhang et al., 2012*). Each toxin is associated with a cognate immunity protein, typically encoded together in an operon, which specifically neutralizes toxicity in the producing cell and in clonemates or close kin that share the locus. The presence of a polymorphic toxin/immunity pair in one strain leads to antagonism toward related strains that do not possess immunity (*Riley and Wertz, 2002*). Examples include contact-dependent growth inhibition (CDI; a type Vb secretion system) (*Aoki et al., 2005*, *Aoki et al., 2010*); modular type IV secretion system (T4SS) (*Souza et al., 2015*), type VI secretion system (T6SS) (*Russell et al., 2011*; *MacIntyre et al., 2010*; *Schwarz et al., 2010*; *Hood et al., 2010*), and type VII secretion system (T7SS) (*Cao et al., 2016*) effectors; as well as the MafB toxins of *Neisseria* (*Jamet and Nassif, 2015a*). The competitive advantages offered by these toxins likely drives positive selection for novel toxin/immunity pairs, which in turn helps to define kin groups through inter-strain antagonism. Mining of prokaryotic genomes revealed that polymorphic toxins are indeed quite prevalent and diverse (*Zhang et al., 2012*). Additionally, many homologous C-terminal toxin domains are shared between distantly related toxin-delivery systems from diverse organisms, suggesting that these toxins have evolved from a common pool of domains (*Zhang et al., 2012*). Currently there is a knowledge gap between the number of toxin domains discovered through bioinformatic analysis and the experimental characterization of their delivery mechanisms (*Jamet and Nassif, 2015b*; *Benz and Meinhart, 2014*). It seems likely that additional, uncharacterized modes of polymorphic toxin delivery remain to be discovered, with each mechanism adapted to the host's particular lifestyle. As mechanisms that promote inter-strain and inter-species conflict, polymorphic toxins appear to play a strong role in the evolution of microbes. For instance, kin discrimination by polymorphic toxins may help maintain cooperation in social organisms such as myxobacteria by promoting local relatedness (*Hamilton, 1964a*; *Vos and Velicer, 2009*). In addition, they likely play an important role in symbiosis (*Hillman and Goodrich-Blair, 2016*) and in population structure within ecological niches such as the soil (*Varivarn et al., 2013*), rhizosphere (*Ma et al., 2014*), and human gut (*Zheng et al., 2015*; *Russell et al., 2014*).

We previously showed that the widely used DK1622 reference strain of *M. xanthus* is killed by ancestral strains when co-cultured on surfaces (*Dey et al., 2016*). A *traA* mutation in either strain abolishes this behavior, indicating that OME is required for antagonism. Further, this antagonism requires a hyper-variable region of the chromosome called Mx-alpha, which is composed of roughly 100 kb of prophage and mobile genetic elements and can be found in multiple copies of imperfect repeats in *M. xanthus* genomes (*Dey et al., 2016*). In ancestral strains that antagonize DK1622, there are three homologous Mx-alpha units apparently arranged in tandem. However, two of these units (~200 kb) were lost by spontaneous deletion during the construction of DK1622 (*Dey et al., 2016*). From these observations, we hypothesized that OME-delivered toxins encoded within the Mx-alpha repeat elements are responsible for antagonism.

Here, we identify the genetic determinant of this antagonism as one of several related, polymorphic, OM lipoprotein toxins that are encoded on Mx-alpha and transferred to target cells by OME. OME between strains that contain different toxins leads to mutual cell death, which establishes territorial barriers between populations. These toxins belong to a large and diverse family found in myxobacteria and display features that make them unique among polymorphic toxin systems. Strikingly, we show that these toxins are serially transferred from cell-to-cell by OME, which results in a potent killing system. Finally, we provide evidence that OME-mediated antagonism contributes to the ecology and evolution of these social microbes.

## Results

### SitA1 is the swarm inhibition toxin

*M. xanthus* inter-strain antagonism related to the presence of Mx-alpha was originally observed as 'swarm inhibition', during which a nonmotile ancestor strain (Mx-alpha$^+$) inhibited the outward swarming of a motile strain (missing two of three Mx-alpha units) during co-culture on agar. This phenomenon is *traA*-dependent and therefore is likely an outcome of OME (*Pathak et al., 2013*) (see *Figure 1*). Swarm inhibition was further demonstrated to be caused by cell death of the motile strain (*Dey et al., 2016*). We first sought to identify the specific genetic determinant on Mx-alpha that was required for antagonism and cell death of the susceptible motile strain. Sequence analysis

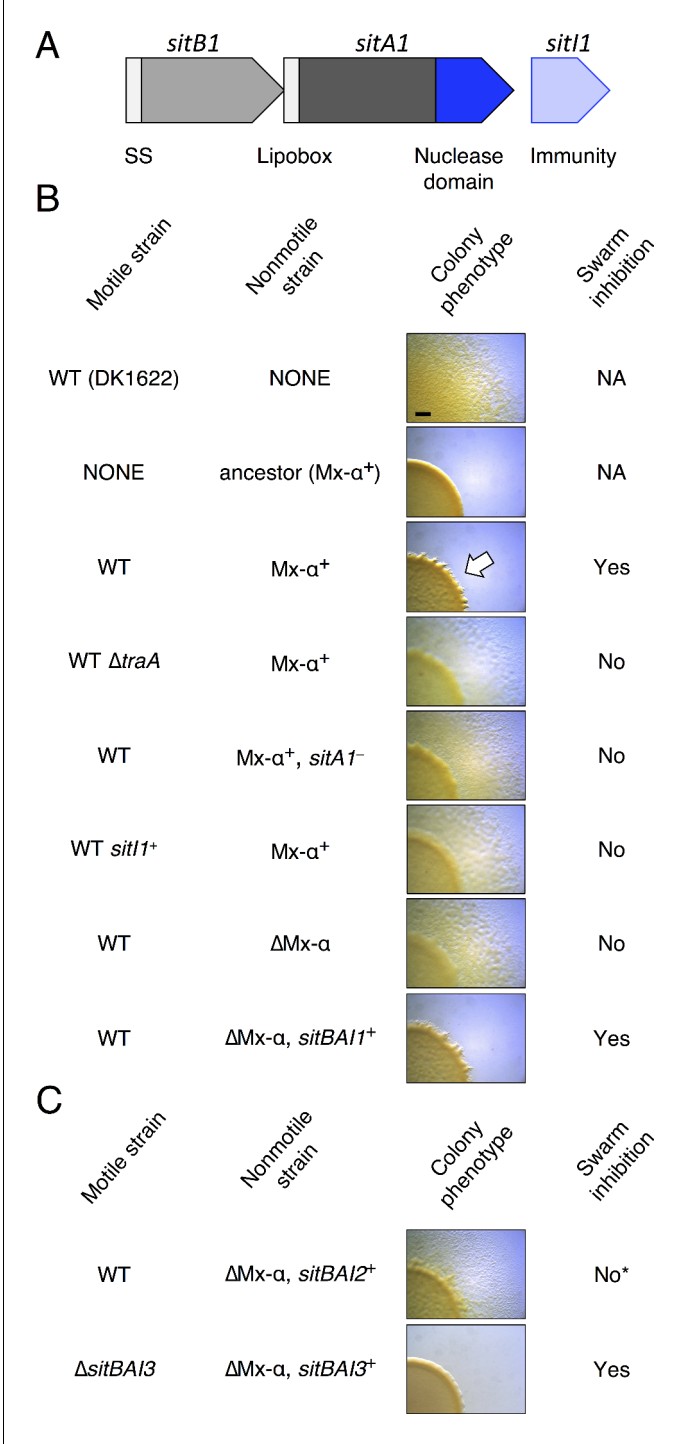

**Figure 1.** SitA1 is the swarm inhibition determinant. (**A**) *sitBAI1* operon found on one of the Mx-alpha elements that was lost from DK1622. SS, signal sequence. (**B**) Swarm inhibition assays with indicated motile and nonmotile strains. White arrow illustrates swarm inhibition with control strains. NA, not applicable. Bar, 1 mm. (**C**) Expression of *sitBAI2* in a non-antagonistic nonmotile background results in modest swarm inhibition (indicated by *) compared to *sitBAI1+* (shown in B). Expression of *sitBAI3* in the non-antagonistic nonmotile background results in complete swarm inhibition of Δ*sitBAI3*. Here and elsewhere see *Supplementary file 2A* for strain details.

of the two Mx-alpha units retained in the ancestral DK101 strain (a.k.a. DZF1) (*Müller et al., 2013*) but lost in DK1622 revealed a candidate toxin gene (MXF1DRAFT_07513), which we have designated *sitA1* for s̲warm i̲nhibition t̲oxin. This gene encodes a predicted lipoprotein that contains a C-terminal nuclease domain with a WHH motif (*Zhang et al., 2011*) (*Figure 1A*). The absence of lysine at the +2 position or alanine at the +7 position relative to the +1 cysteine in the N-terminal lipobox suggests that this protein is localized to the OM (*Bhat et al., 2011*) (see *Supplementary file 1*). Given that OME efficiently transfers OM lipoproteins between cells (*Wei et al., 2011*), this open reading frame (ORF) represents a promising candidate for the antagonistic determinant. Immediately downstream of *sitA1* is a gene (*sitI1*) that shows homology to the SUKH-family of immunity proteins commonly found in polymorphic toxin systems (*Zhang et al., 2011*) (*Figure 1A*). Upstream of *sitA1* is a hypothetical gene (*sitB1*) of unknown function. The *sitB1* ORF overlaps with *sitA1* by 11 base pairs, suggesting that the genes form an operon and function together (*Figure 1A*).

To test if *sitA1* is the swarm inhibition toxin, we used the swarm inhibition assay as a readout for the contribution of *sitA* or *sitI* toward cell death of the susceptible motile strain during co-culture with the antagonistic, nonmotile ancestral strain. In this assay, cell death of the susceptible strain results in no cells visibly escaping the mixed culture spot. In contrast, abrogation of cell death results in the appearance of the motile strain moving outward from the colony co-culture. As shown previously (*Pathak et al., 2013*; *Dey et al., 2016*; *Dey and Wall, 2014*), nonmotile ancestral cells inhibited the motility of DK1622, but not the DK1622 Δ*traA* strain (*Figure 1B*, rows 1–4). Importantly, nonmotile ancestors carrying a *sitA1* mutation did not inhibit DK1622 (*Figure 1B*, row 5). Further, expression of *sitI1* in motile DK1622 cells also prevented antagonism (*Figure 1B*, row 6), consistent with the prediction that *sitI* encodes an immunity protein that neutralizes SitA1. To test whether *sitBAI1* is sufficient to convert non-antagonistic cells into killers, we expressed the gene cassette in a DK1622-derived nonmotile strain, which lacks two Mx-alpha units (see *Figure 2A*), and does not cause swarm inhibition. As predicted, ectopic *sitBAI1* expression allowed the nonmotile ΔMx-alpha cells to inhibit DK1622, thus recapitulating the antagonistic phenotype exhibited by the nonmotile ancestor strain (*Figure 1B*, rows 7–8). These combined results suggest that *sitBAI1* may function as a toxin/immunity system responsible for the antagonistic behaviors previously described (*Pathak et al., 2013*; *Dey et al., 2016*; *Dey and Wall, 2014*). Therefore, the loss of two Mx-alpha units, and thus the *sitBAI1* operon (*Figure 2A*), during the construction of DK1622 from an ancestral DK101 strain explains why the latter strain antagonizes the former.

## DK1622 ancestors contain three functional *sitBAI* toxin/immunity cassettes

The three tandem Mx-alpha units in the ancestor strain are related and contain different alleles of >80 genes. This region therefore represents a rare bacterial polyploid element – that is, it contains three Mx-alpha prophage genomes with divergent gene allele sets (*Dey et al., 2016*). Inspection of the other two Mx-alpha units revealed additional putative *sitBAI* operons. The second Mx-alpha unit (absent from DK1622) carries *sitA2* (MXF1DRAFT_07313), which encodes a putative lipoprotein with clear homology to the N-terminal region of SitA1, though the C-terminal domains are unrelated (*Figure 2A*). The *sitA2* gene is flanked by a *sitB1* homolog, *sitB2*, and a downstream candidate immunity gene, *sitI2* (*Figure 2A*). The third Mx-alpha unit, which is shared between the ancestral strain and DK1622, also appears to contain a *sitBAI* operon. Although the *sitA3* gene (MXF1DRAFT_05864 or MXAN_1899) has low sequence homology with *sitA1* and *sitA2*, the three genes nevertheless share several key features: (1) *sitA3* is preceded by *sitB3*, which is homologous to *sitB1* and *sitB2*, (2) *sitA3* occupies a similar position within its Mx-alpha unit as the other *sitA* genes, (3) *sitA3* encodes an OM lipoprotein signal sequence, (4) *sitA3* encodes a predicted C-terminal tRNase toxin domain, and (5) *sitA3* is adjacent to a downstream putative immunity gene, *sitI3* (*Figure 2A*). This analysis suggests that the three Mx-alpha units each contain distinct *sitBAI* toxin/immunity operons.

To determine whether SitA lipoproteins function as toxins, we expressed each *sitBAI* cassette in DK1622 and tested the competitive fitness of the resulting inhibitor strains against parental DK1622 target cells that lack the corresponding *sitI* gene. Target strains were labeled with fluorescent markers and co-cultured with inhibitor strains on agar for 24 hr. Competition outcomes were assessed by competitive index, which is the ratio of target cells to toxin-producing inhibitor cells at 24 hr relative to the starting ratio. For example, a competitive index of 0.01 indicates that the ratio

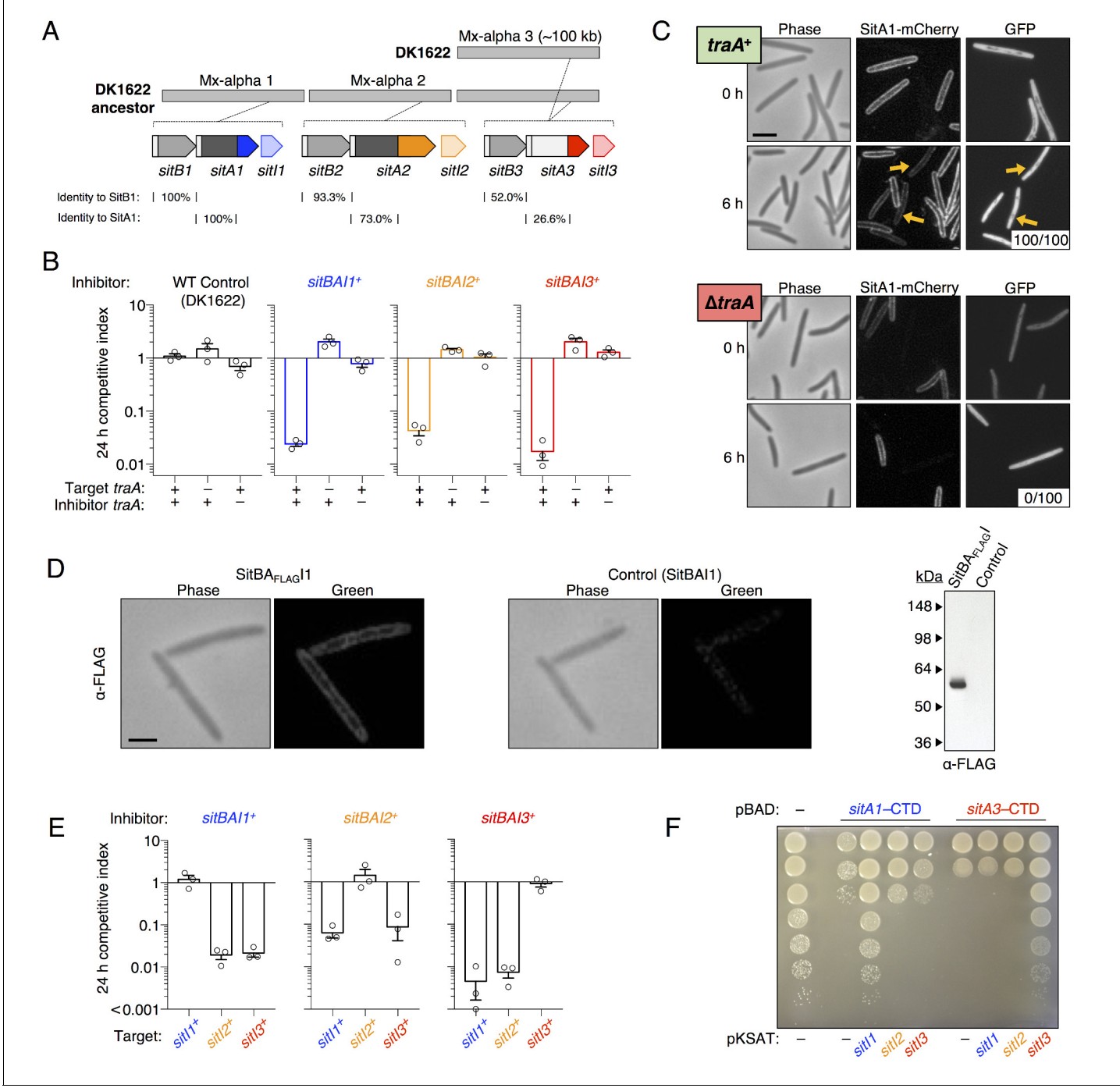

**Figure 2.** SitA polymorphic toxins found on Mx-alpha units are delivered by OME. (**A**) Strain DK101 (the ancestor of DK1622) carries three Mx-alpha repeats, whereas DK1622 retains only one copy. Each Mx-alpha unit contains a unique *sitBAI* cassette. SitB proteins contain type I signal sequences (white boxes) whereas SitA proteins contain type II signal sequences (white boxes) with a lipobox and C-terminal toxin domains. The relative sequence identities are shown. (**B**) Competition outcomes when inhibitor strains each expressing one of three *sitBAI* cassettes were competed against susceptible target strains that lack the corresponding *sitBAI* cassette. Mock-inhibitor control is shown at left (WT vs. WT). See text for the calculation of competitive index. Strain genotypes ('–', *traA* deletion) are shown below histograms and further strain details provided in ***Supplementary file 2A***. (**C**) Cells harvested from an agar co-culture of a strain expressing a SitA1-mCherry fusion with a GFP-labeled target at 0 and 6 hr. GFP targets are *traA⁺* in the top panel and Δ*traA* in the bottom panel. Yellow arrows indicate two examples of GFP cells that have acquired the mCherry reporter. Boxes represent the number of mCherry positive GFP cells out of 100. Bar, 5 μm. (**D**) Fixed-cell immunofluorescence of C-terminal FLAG-tagged SitA1 and untagged control. Bar, 2.5 μm. Immunoblot of protein isolated from the same strains (right). SitA_FLAG predicted size is 62.6 kDa. (**E**) Competition outcomes when inhibitor expresses one of the three *sitBAI* cassettes and the target strains express one of the three *sitI* genes. Data points at <0.001

*Figure 2 continued on next page*

*Figure 2 continued*

indicate that no target cells remained. (**F**) *E. coli* MG1655 plating efficacy when equal number of cells were 10-fold serially diluted, spotted onto arabinose-supplemented agar and incubated overnight. Strains express either SitA1 or SitA3 C-terminal toxin domain (CTD) from a pBAD plasmid either in the absence ('–', empty vector) or presence of the indicated *sitI* genes expressed constitutively from a separate plasmid (pKSAT). This image is representative of three biological replicates. In this figure and the figures below, error bars represent standard error of the mean from at least three independent experiments.

The following figure supplements are available for figure 2:

**Figure supplement 1.** Morphology of SitA-poisoned target cells.

**Figure supplement 2.** SitA-CTD expression in *M. xanthus* is toxic.

**Figure supplement 3.** Heterologous *sitAI* cassettes from *M. fulvus* HW-1 are active in DK1622.

of target cells to inhibitor cells decreased 100-fold. In all instances, *sitBAI*-expressing inhibitor cells significantly outcompeted target cells, whereas the mock-inhibitor did not (**Figure 2B**). Delivery of SitA1 and SitA2 over a 24 hr period induced filamentation and lysis of target cells, whereas SitA3 induced rounding and lysis of target cells (**Figure 2—figure supplement 1**). Furthermore, a Δ*traA* mutation in either the target or inhibitor strain abolished the inhibitor's competitive advantage (**Figure 2B**). We note that assessing competitive index by microscopy gives a quick and reproducible metric of one strain's ability to outcompete another, but does not capture the full dynamic range of competition because many enumerated target cells have severe morphological abnormalities and are likely not viable at the 24 hr time point. However, microscopy allows competitive indices to be determined for these otherwise WT strains, which are not easily amenable to enumeration as colony forming units (CFU) because they form extraordinarily cohesive biofilms in isogenic co-cultures (dependent on type IV pili). These results show that SitA lipoproteins provide a competitive advantage, conferring the ability to kill and/or inhibit the growth of competitors in a TraA-dependent manner.

To examine SitA localization, we generated an mCherry reporter that carries the N-terminal lipobox from SitA1. Cells expressing this fusion have membrane-localized fluorescence as expected for a lipoprotein (**Figure 2C**). The TraA-dependent function of SitA shown in **Figure 2B** suggests that the protein is delivered by OME. Therefore, we tested whether the reporter fusion is transferred between cells. We co-cultured the reporter strain with a target strain expressing cytoplasmic GFP (which is not exchanged [**Wei et al., 2011**]) and microscopically assayed for transfer of the reporter. At 6 hr of co-culture, we observed the mCherry signal present in the cell envelope of the GFP target strain, indicating cell-to-cell transfer of the SitA1-mCherry reporter (**Figure 2C**, upper panel). Deletion of *traA* in the GFP target strain prevented the acquisition of mCherry signal (**Figure 2C**, lower panel), recapitulating our earlier findings that OM-localized reporters are exchanged between cells in a TraA/B-dependent manner (**Nudleman et al., 2005**; **Wei et al., 2011**; **Pathak et al., 2012**). Because inner membrane lipoproteins are not transferred during OME (**Wei et al., 2011**), these data also suggest that the lipobox directs SitA1 to the OM. We confirmed that full-length SitA1 localizes to the cell envelope using immunofluorescence microscopy to detect FLAG epitope-tagged SitA1 in formaldehyde-fixed cells (**Figure 2D**). Taken together, these results demonstrate that SitA1 resides in the OM and is transferred cell-to-cell by OME.

The fact that *sitI1* expression protects WT DK1622 cells from swarm inhibition suggests that this gene encodes an immunity protein that neutralizes SitA1 toxicity. To determine whether SitI proteins block SitA-mediated growth inhibition, we expressed each *sitI* allele individually in DK1622 Δ*sitBAI3* cells and co-cultured the resulting strains with strains that express each of the three *sitBAI* cassettes. For each competition, only strains that express the cognate *sitI* were protected from growth inhibition (**Figure 2E**), consistent with immunity function.

Immunity proteins typically interact with the C-terminal domain of polymorphic toxins (**Zhang et al., 2012**; **Poole et al., 2011**). To test whether this was true for SitA, we expressed the predicted C-terminal toxin domains (CTD) of each SitA toxin in *E. coli* MG1655 under the inducible $P_{BAD}$ promoter. Expression of SitA1-CTD and SitA3-CTD blocked growth, but co-expression of

cognate *sitI* from a second plasmid restored cell growth (*Figure 2F*). These results confirm that SitI proteins specifically neutralize cognate SitA toxins. In addition, because the SitA-CTD constructs lack secretion signal sequences, these data show that the domains exert their toxic effects in the cytoplasm. We also tested SitA2-CTD expression constructs, but none inhibited *E. coli* MG1655 growth. Because SitA2-mediated inhibition is obvious in *M. xanthus* competition co-cultures (*Figure 2B and E*), we tested the SitA-CTD expression constructs in *M. xanthus* and found that each inhibited cell growth (*Figure 2—figure supplement 2A*). Therefore, SitA2-CTD is indeed toxic when expressed in the cytoplasm of *M. xanthus.*

Given that *sitBAI2* expression confers a significant advantage in competition co-culture (see *Figure 2B*), it is unclear why swarm inhibition was not observed with the *sitA1⁻* nonmotile ancestral strain (see *Figure 1B*, row 5), considering these cells should still deploy SitA2 and that DK1622 lacks the SitI2 immunity protein. Therefore, we tested whether nonmotile ∆Mx-alpha cells that ectopically express *sitBAI2* are able to inhibit DK1622 swarming. In agreement with the prior result, we found that DK1622 motility was only partially inhibited by the *sitBAI2*-expressing strain (*Figure 1C*, row 1). This result confirms that SitA2 contributes to the swarm inhibition phenotype, but is not sufficiently potent by itself to block outward swarming of DK1622. Together, these results indicate that SitA1 is the major swarm-inhibition toxin. We note that SitA3 does not contribute to the originally observed swarm inhibition phenotype because both ancestor and DK1622 strains contain the *sitBAI3* operon (see *Figure 2A*). However, we found that a nonmotile strain expressing *sitBAI3* fully inhibits the motility of a DK1622 ∆*sitBAI3* strain that lacks the *sitI3* immunity gene (*Figure 1C*, row 2).

## SitA C-terminal domains are nuclease toxins

Homologous CTDs are often associated with different toxin delivery systems from phylogenetically distant bacteria (*Zhang et al., 2012*). SitA3-CTD is homologous to a previously characterized tRNase domain found at the C-terminus of CdiA from *Burkholderia pseudomallei* 1026b (*Morse et al., 2012*; *Nikolakakis et al., 2012*) and an orphan CdiA-CTD encoded by *Yersinia pseudotuberculosis* YPIII (*Figure 3—figure supplement 1*). To determine whether SitA3-CTD also has tRNase activity, we expressed the toxin in *E. coli* and compared its activity to the CDI toxins. Induction of SitA3-CTD expression inhibited cell growth in the same manner as the CdiA-CTD toxins (*Figure 3A*). Examination of tRNA from SitA3-CTD intoxicated cells revealed cleavage of $tRNA_{UGC}^{Ala}$, similar to the specific tRNase activity of the *B. pseudomallei* toxin (*Figure 3A*).

Next, we investigated the toxic activities of SitA1-CTD and SitA2-CTD. We had previously observed that SitA1 induces cell filamentation and loss of DAPI staining in *M. xanthus* target cells, which is consistent with DNase activity mediated by the predicted Colicin-DNase domain (Pfam 12639, E = 6.7 e-21) containing the WHH motif (*Zhang et al., 2011*). To test this, we induced SitA1-CTD expression in *E. coli* and found that cells became filamentous and had reduced DAPI stain signal (*Figure 3B*). By contrast, *E. coli* cells that were intoxicated by SitA3-CTD retained DAPI staining, though their nucleoids became more compact (*Figure 3B*). Together, these results suggest that SitA1-CTD has DNase activity. HMM-HMM comparison (HHpred [*Söding et al., 2005*]) of C-terminal residues 699–783 of SitA2 revealed distant homology to another CdiA-CTD from *Y. pseudotuberculosis* YPIII (locus tag, Ga0077885_11586), which was previously characterized as a DNase (*Morse et al., 2015*). To examine this possibility, we expressed each SitA-CTD in *M. xanthus* under the control of an IPTG-inducible promoter. Expression of SitA2-CTD in *M. xanthus* resulted in cell filamentation and reduced DAPI staining (*Figure 2—figure supplement 2B*), suggesting that SitA2-CTD degrades DNA. Expression of SitA1-CTD and SitA3-CTD in the cytoplasm of *M. xanthus* cells yielded similar results as when they were expressed in *E. coli* (*Figure 2—figure supplement 2B*), although the DAPI signal from SitA3-CTD expressing cells was brighter than the control and many cells contained two distinct nucleoids (*Figure 2—figure supplement 2B*), suggesting a block in cell division.

## Polymorphic SitA toxins are conserved in myxobacteria

To determine the phylogenetic distribution of SitA toxins, we conducted a BLAST search using the N-terminal domains of SitA1/2 (which are homologous) and SitA3 as query sequences. This search recovered >100 *sitA* orthologs that are common in the Myxococcales (*Supplementary file 1*). More

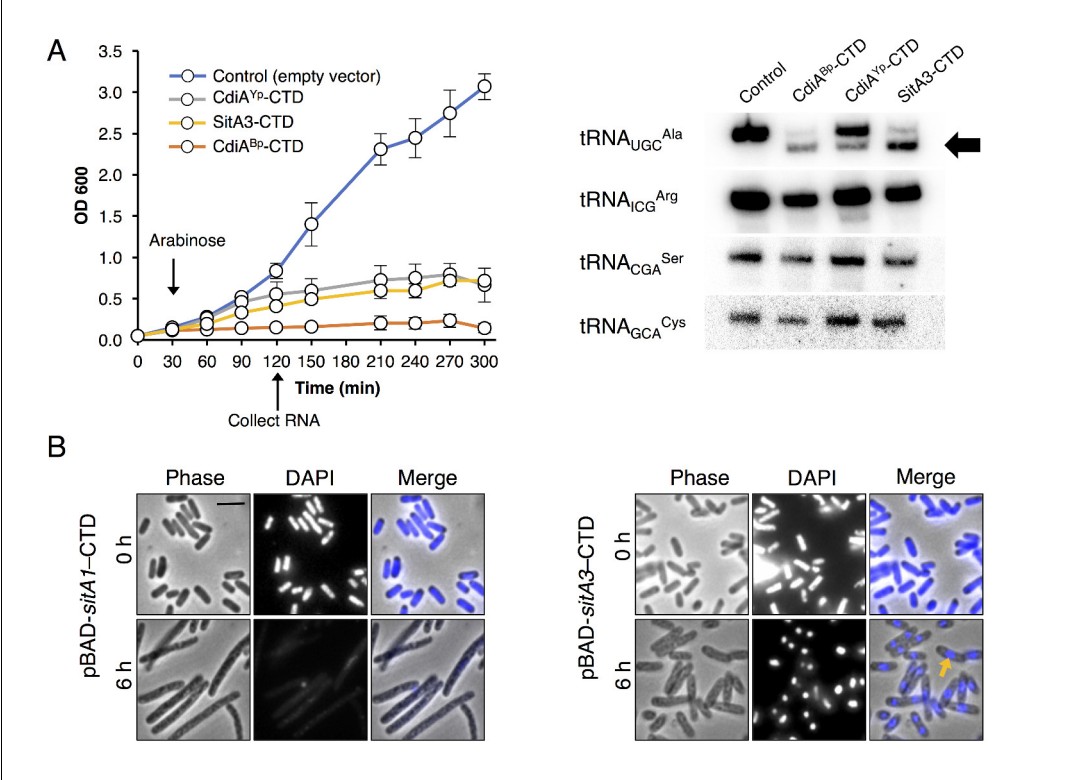

**Figure 3.** Toxic function of SitA1 and SitA3 CTDs. (A) SitA3-CTD is a toxic tRNase. Expression of the indicated CTDs was induced with arabinose in *E. coli*, and cell growth monitored by measuring the optical density of the cultures at 600 nm (OD$_{600}$). OD$_{600}$ values are reported as the average ± standard error for three independent experiments (left). RNA was isolated after 90 min of toxin expression and analyzed by Northern blot hybridization using probes to the indicated tRNAs (right). The arrow indicates cleaved tRNA$_{UGC}^{Ala}$. (B) SitA1-CTD has DNase activity. *E. coli* cells were stained with DAPI at 0 hr and after 6 hr of toxin expression. Cells expressing *sitA1-CTD* became filamentous and lost DAPI staining at 6 hr (left). In contrast, *sitA3-CTD* expressing cells retained DAPI staining (right), though their nucleoids became compacted (yellow arrow). Bar, 5 μm.

The following figure supplement is available for figure 3:

**Figure supplement 1.** Alignment of SitA3-CTD with CdiA-CTD tRNase toxins.

sensitive search algorithms such as HMMER (*Finn et al., 2011*) failed to identify significant homologs outside of the Myxococcales. Consistent with the finding that SitA is delivered through OME, all orthologs contain lipoboxes and are only found in species that contain *traAB*. Moreover, SitA C-terminal domains are variable and typically show homology to nuclease domains when subjected to HMM-HMM comparison using HHpred (*Supplementary file 3*). Interestingly, many *sitA* genes are not linked to upstream *sitB* orthologs, particularly in species that are distantly related to *M. xanthus* (*Supplementary file 1*). This suggests that SitB may not be required for SitA function, or perhaps that SitB proteins, encoded at unlinked loci, function promiscuously between multiple SitA proteins. Notably, some *sitA* loci are found outside of Mx-alpha-like elements, however, these genes are typically adjacent to other mobile genetic elements. To test cross-genotype compatibility of SitA orthologs, we cloned two *sitA* gene cassettes from *Myxococcus fulvus* HW1 for heterologous expression in *M. xanthus*. One of these operons (*sitAI3^Mf1*) does not contain a *sitB* gene. As predicted, *M. xanthus* cells that express heterologous *sitAI3^Mf1* or *sitBAI1^Mf1* outcompeted the parental strain in a *traA*-dependent manner (*Figure 2—figure supplement 3*). These results indicate that SitA toxin delivery is not limited by its specific species/strain of origin, and that the systems are functional after horizontal gene transfer (HGT) of a minimal set of components (*sitAI*).

## SitA toxins define barriers to social compatibility

Our results show that *sitBAI*-expressing cells inhibit OME-compatible strains that lack a cognate SitI immunity protein. We hypothesized that this antagonism should be sufficient to mediate territorial exclusion. Territorial exclusion promotes physical segregation of nonself organisms (*Gibbs and Greenberg, 2011*), which in turn drives both diversification and maintenance of cooperation (*Papke and Ward, 2004*; *Velicer and Vos, 2009*). Territorial exclusion between wild *M. xanthus* genotypes, including those that are closely related and that are isolated in close proximity to one another has been well studied, but the specific determinants underlying this behavior are unknown (*Vos and Velicer, 2009*; *Rendueles et al., 2015*; *Vos and Velicer, 2006*; *Wielgoss et al., 2016*). We tested whether SitA is sufficient for territorial exclusion by conducting colony merger assays with every combination of DK1622 strains expressing one of the five described *sitAI* alleles. In these assays, two liquid cultures are spotted next to one another on agar, and the colonies are allowed to swarm toward each other. If the converging swarms merge, then the strains are considered compatible. For each combination, the expression of different *sitAI* cassettes resulted in dramatic lines of demarcation between the two strains (*Figure 4A*), and the formation of these demarcation zones was *traA*-dependent (*Figure 4B*). These results show that otherwise isogenic strains are rendered socially incompatible and geographically isolated by the acquisition of a single *sitAI* cassette.

## SitA toxins are infectious

We previously found that swarm inhibition occurs efficiently even when motile cells outnumber the antagonistic nonmotile strain 40 to 1 (*Dey and Wall, 2014*). This observation suggests that an individual SitA-expressing cell can inhibit many targets. To further explore this phenomenon, we quantified viable target cells in a series of competition co-cultures in which the ratio of SitA producing cells to targets was progressively decreased by factors of 10. To facilitate CFU enumeration in these experiments, we used $\Delta pilA$ cells, which are unable form type IV pili-dependent biofilms. We found that target cell CFU were reduced approximately $10^6$-fold when the strains were mixed at a 1:1 ratio (*Figure 5A*). The higher degree of killing reported here (compared to competitive index in *Figure 2B*) provides a clearer understanding of the killing efficiency because the CFU assay measures a broader dynamic range of viable cell number. Remarkably, the SitA-producing strain still reduced target cell viability $>10^4$-fold in co-cultures seeded at a 1:1000 ratio of inhibitors to target cells (*Figure 5A*). These observations imply that each inhibitor cell intoxicates several thousand target cells during co-culture. In one explanation we hypothesized that OME delivery allows a series of SitA transfer events from one target cell to other cells. We consider this serial transfer mechanism plausible if translocation of all toxin molecules from the target cell OM to the cytoplasm is not completed before subsequent OME events occur. This model predicts that SitA toxins could spread through the population like an infectious agent, intoxicating target cells that never made direct contact with the producer. To test this hypothesis, we conducted three-strain competitions with (1) a *sitBAI1* inhibitor strain that contains *M. fulvus traAB* (*traAB*^Mf) as its only *traAB* alleles, (2) a susceptible target strain that contains *M. xanthus traAB*^DK1622 alleles and thus is incompatible for OME with the inhibitor, and (3) a susceptible intermediary strain that carries both *traAB*^Mf and *traAB*^DK1622 (*Figure 5B*). If serial toxin transfer occurs, the *traAB* merodiploid strain should act as an intermediary carrier/conduit to deliver toxin to *traAB*^DK1622 targets (*Figure 5B*). As a control, we first showed that *traAB*^Mf inhibitors do not inhibit *traAB*^DK1622 targets (*Figure 5C*), consistent with the incompatibility of their *traAB* alleles. Importantly, inclusion of intermediary cells, which are inhibited (*Figure 5D*), also resulted in the inhibition of *traAB*^DK1622 target cells (*Figure 5E*). As expected, the intermediary strain (which lacks *sitBAI1*) did not inhibit *traAB*^DK1622 targets in co-cultures containing only those two strains (*Figure 5F*). To exclude a SitA-independent mechanism of target cell inhibition, we conducted the same three-strain competition, but provided *traAB*^DK1622 targets with the *sitI1* immunity gene. In this latter co-culture, the intermediary strain was inhibited, but *traAB*^DK1622 targets were not (*Figure 5G*). Finally, we tested an intermediary strain that lacks *traA* and found that neither intermediary nor target cells were inhibited during co-culture (*Figure 5H*).

To directly visualize serial transfer, we co-cultured *traAB*^Mf cells that express the SitA1-mCherry fusion (described in *Figure 2*) with *traAB* merodiploid intermediary cells that express cytoplasmic tdTomato, and GFP-labeled, *traAB*^DK1622 target cells. Microscopic examination of target cells at 6 hr revealed that all had acquired mCherry fluorescence, with the signal localized to the cell envelope

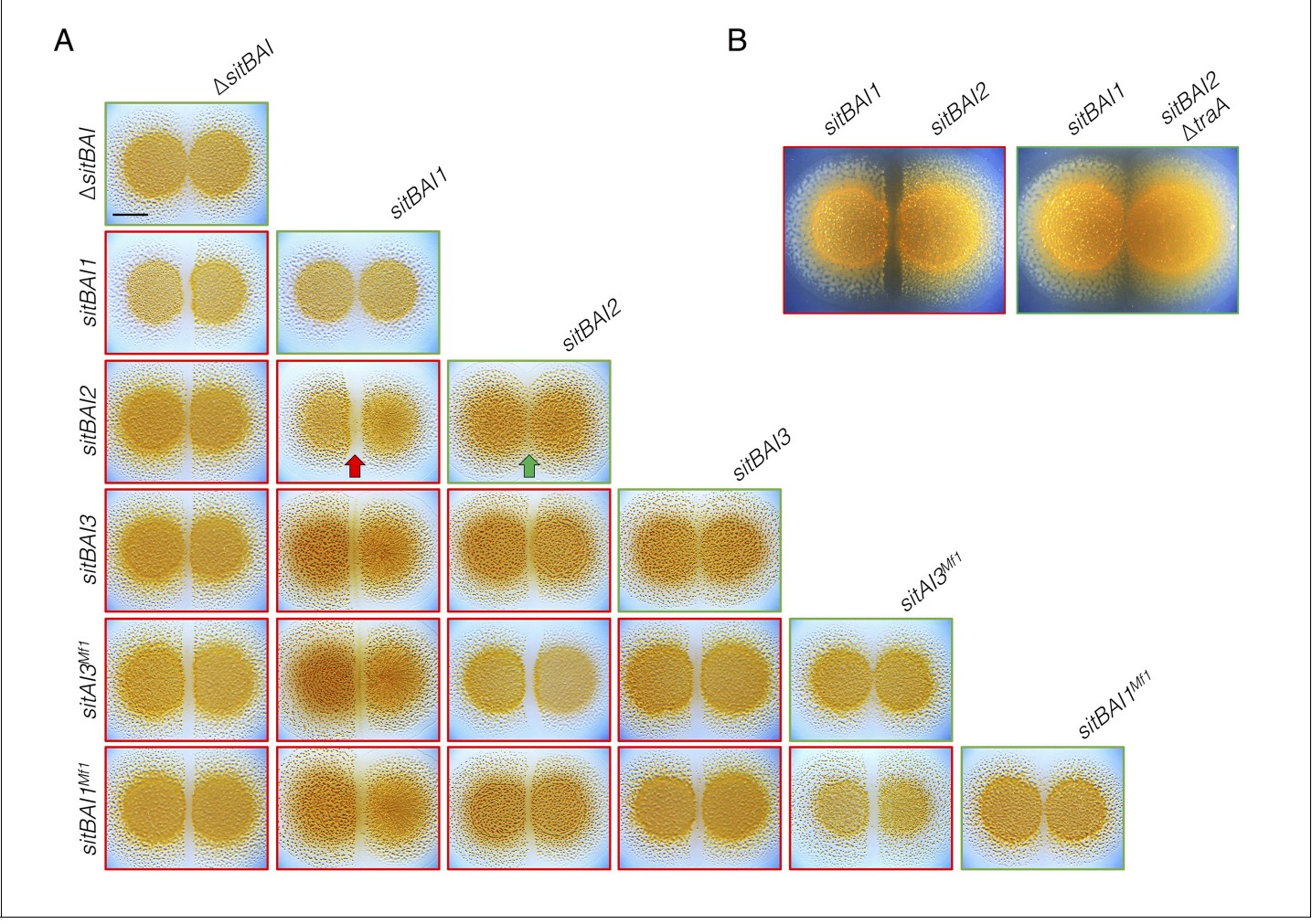

**Figure 4.** *sitAI* alleles determine the social compatibility of *M. xanthus* swarms. (**A**) *M. xanthus* colonies expressing identical *sitAI* cassettes merge (as illustrated by the green arrow) when spotted adjacent to one another (top of each column). Strains that express different *sitAI* cassettes form demarcation zones between colonies (illustrated by the red arrow). Labels on the left indicate toxin expressed by left colony, while top labels indicate toxin expressed by colony on the right. Green borders indicate colony merging and red indicates demarcation. (**B**) Demarcation zone formation is *traA*-dependent. Bar, 5 mm.

(*Figure 5I*, left panels). By contrast, no SitA1-mCherry transfer was detected when the intermediary was absent from the co-culture (*Figure 5I*, right panels). Although these results are consistent with serial transfer of SitA, it is also possible that target cells acquire TraA^Mf and/or inhibitor cells acquire TraA^DK1622 by OME-dependent exchange of TraA with the intermediary strain, which would then allow direct transfer of SitA between inhibitor and target cells. To test if TraA is transferred during OME, we used a TraA-mCherry fusion to monitor transfer of TraA. The TraA fusion promoted efficient transfer of an OM sfGFP reporter (*Figure 5—figure supplement 1A*), demonstrating that it is functional to catalyze OME. However, TraA-mCherry itself was not transferred (*Figure 5—figure supplement 1B*). One explanation for why TraA does not transfer is that it interacts with TraB, which contains an OmpA domain known to bind the cell wall. For this reason, we suspect TraA is anchored to the cell envelope and is unable to transfer (*Cao and Wall, 2017*). Taken together, these results indicate that SitA1 can be transferred from the initial target to secondary recipients, supporting a model in which SitA acts like an infectious agent that disseminates through a population by OME.

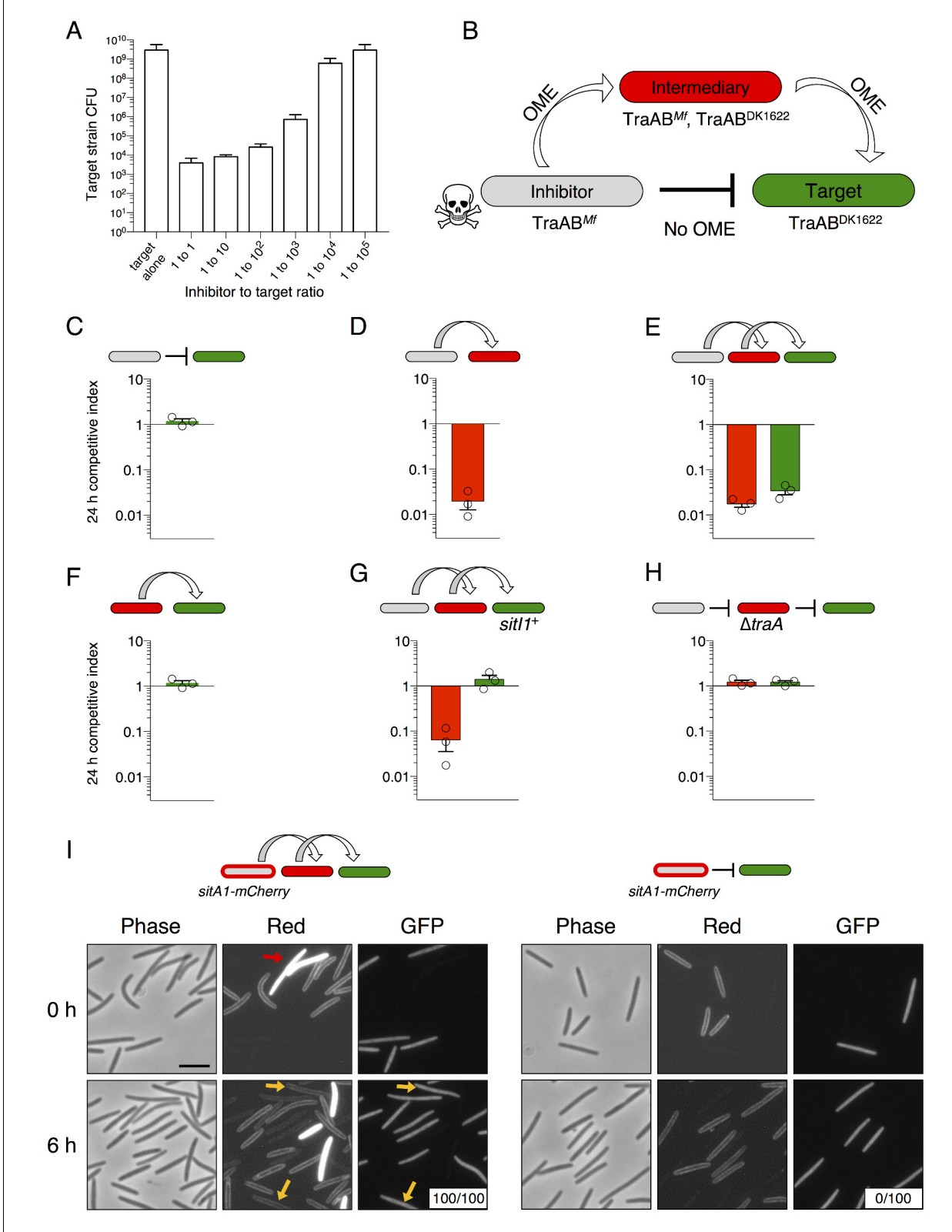

**Figure 5.** SitA toxins are serially transferred by OME. (**A**) Viable cells (CFU) of a target population as a function of inhibitor to target cell ratio quantifies the efficiency of SitA1 and OME delivery. Strains were co-cultured on agar for 48 hr at indicated ratios before determining CFU of the marked (Km$^r$) target strain. (**B**) Experimental design to test serial transmissibility of SitA1. The grey cell produces the SitA1 toxin and contains *traAB$^{Mf}$* alleles. The target cells (green) are susceptible, but carry incompatible *traAB$^{DK1622}$* alleles that preclude OME with inhibitors. Intermediary cells (red) express both

*Figure 5 continued on next page*

*Figure 5 continued*

*traAB* alleles. (C–F) Competitive indices of intermediary (red) and target (green) strains from two- and three-strain co-cultures. (G) Three-strain competition when the target strain expresses SitI1. (H) Three strain competition when the intermediary strain is ΔtraA. Competition outcomes were determined at 24 hr by fluorescent microscopy. Competitive index was calculated relative to the inhibitor (C–E, G–H) or relative to intermediary strain (F). Starting ratio was 1:5:5 inhibitor to intermediary to target. (I) Serial transfer of the SitA1-mCherry fusion. The left panel shows a 10:1:1 mixture of *sitA1*-mCherry cells to intermediary to target visualized at 0 and 6 hr. Red arrow indicates a representative example of an intermediary cell that expresses cytoplasmic tdTomato (which does not transfer). Yellow arrows indicate GFP-labeled target cells that have acquired an OM-localized mCherry signal at 6 hr. Boxes represent the number of mCherry positive GFP cells out of 100. Right panel: otherwise identical experiment omitting the intermediary strain. Bar, 5 μm.

The following figure supplement is available for figure 5:

**Figure supplement 1.** TraA is not transferred during OME.

## SitB is an accessory protein that contributes to SitA function

In *M. xanthus* and its close relatives, *sitA* is typically accompanied by an overlapping *sitB* cistron. In the case of DK101 the genes overlap by 11 bp in all three *sitBAI* cassettes. SitB shows no significant homology to other proteins or domains using HMMER or HHpred, though it does contain a type I signal sequence (SignalP 4.1 [*Petersen et al., 2011*]). I-TASSER (*Zhang, 2008*) predicts that SitB adopts a transmembrane β-barrel structure characteristic of OM proteins. To examine the role of SitB1, we tested the activity of inhibitor cells that express either *sitBAI1* or *sitAI1* (cells lack *sitB1*) against a susceptible target strain. The inhibitors in these experiments also carried a Δ*sitB3* mutation to eliminate the possibility of promiscuous interactions between SitB3 and SitA1. At a 1:1 (inhibitor to target) ratio, *sitAI* inhibitors had less of an advantage against targets than *sitBAI* inhibitors, but still retained activity compared to the mock inhibitor control (*Figure 6A*, left). The *sitAI* inhibitors were less effective at a 1:10 ratio, and at 1:100 were indistinguishable from mock inhibitors (*Figure 6A*). In contrast, *sitBAI1* inhibitors were equally effective at outcompeting the target strain at each of the three ratios (*Figure 6A*). Thus, SitB1 contributes significantly to SitA1-mediated inhibition.

Progressive loss of function at increasing target to inhibitor ratios with the *sitAI* inhibitors could indicate defects in serial toxin transfer compared to *sitBAI* inhibitors. Therefore, we tested serial transfer using three-strain co-cultures as described in *Figure 5B*. To improve the sensitivity of this

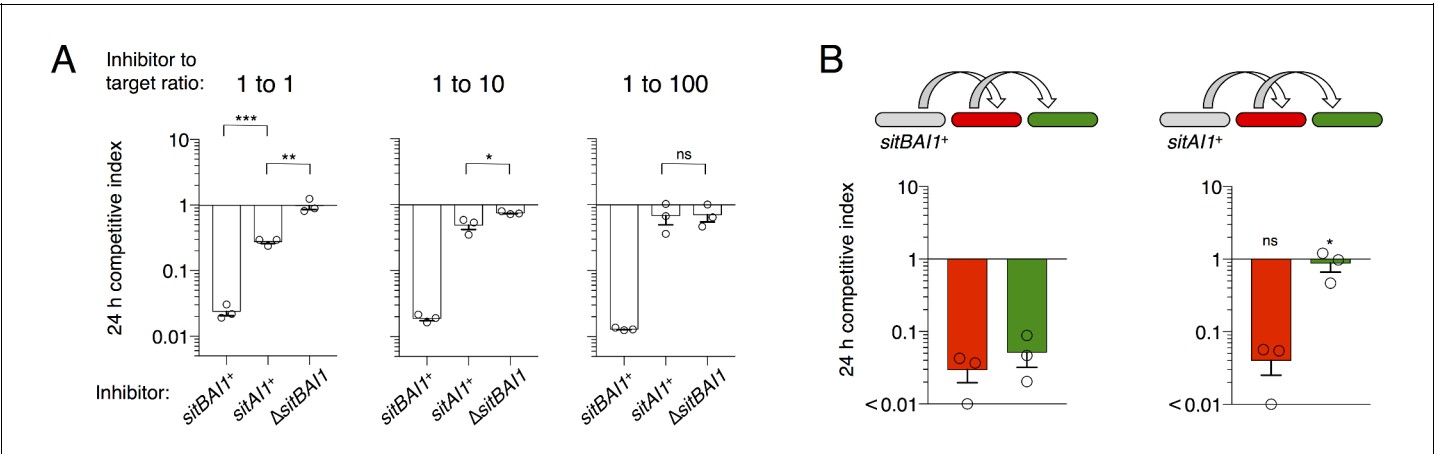

**Figure 6.** SitB contributes to SitA function and serial transfer. (A) The indicated SitA1 inhibitor strains were co-cultured with target cells at three different inhibitor to target ratios. Competitive index was measured at 24 hr by counting the ratios of fluorescently marked cells. Asterisks indicate level of statistical significance, ns = not significant. P-values of indicated comparisons from left to right: 0.0002, 0.006, 0.0257, 0.9359. (B) Serial transfer was monitored as in *Figure 5* using *sitBAI* or *sitAI* inhibitors that express *traAB^{Mf}*. Co-cultures were seeded at a 10:1:1 ratio of inhibitor to intermediary to target strains. Significance indicators refer to comparisons between the inhibitor strains. P-values from left to right: 0.6341, 0.0193.

assay, we increased the inhibitor to intermediary to target strain ratio to 10:1:1 to compensate for the inhibition defect of *sitAI1* cells. As observed in *Figure 5*, *sitBAI1* cells outcompeted both the intermediary and target strains (*Figure 6B*). Because of the high inhibitor cell ratio, the *sitAI1* cells outcompeted the intermediary strain to a similar extent as *sitBAI* cells, but importantly, *sitAI* cells had little to no effect on target cells (*Figure 6B*). We extended the experiment to 48 hr and also performed experiments with SitA-resistant intermediary cells, to increase the number of conduit cells, but again we did not observe SitAI-mediated antagonism of target cells. These results support the hypothesis that SitB promotes SitA transfer, including the serial transfer from primary to secondary target cells. Although SitB1 clearly contributes to SitA1-mediated inhibition, it is not strictly required, which may explain why many myxobacterial *sitA* genes are not linked to *sitB*. Finally, these results are congruent with our above conclusion that TraA is not transferred, because if it was, then direct transfer of SitA1 would occur between *sitAI* inhibitors and the target strain. However, this did not occur because the target strain was not inhibited.

## OME and SitA are critical for competitive fitness within TraA recognition groups

*M. xanthus* uses multiple inhibitory mechanisms to antagonize non-kin. However, because SitA toxins are serially transferred between cells, we hypothesized that they should be powerful determinants of competitive outcomes during inter-genotype conflict. To test this hypothesis, we quantified the contribution of TraA and SitA to competitive outcomes in co-cultures of DK1622 with wild *M. xanthus* soil isolates. Isolates A66 and A88 (from Tübingen, Germany) (*Vos and Velicer, 2006*) and DK801 (from California, USA) (*Martin et al., 1978*) contain *traA* alleles in the same recognition group as DK1622 (originally isolated from Iowa, USA) (*Pathak et al., 2013*; *Dey et al., 2016*). We compared the fitness outcomes of WT, Δ*traA*, and Δ*sitBAI3* genotypes when co-cultured with these environmental isolates by monitoring the ratio of fluorescently labeled DK1622-derived cells to isolate cells at 4, 8 and 24 hr. As a second metric, we enumerated CFU of the DK1622-derived strains at the 24 hr time point. In every case, mutant strains that cannot deploy SitA3 had dramatically decreased competitive fitness outcomes and viability compared to WT (*Figure 7A*). Remarkably, against all three isolates, the presence of *traA* was the determining factor in which strain prevailed, demonstrating up to a $10^6$-fold swing in strain ratio (vs. A66) and a near $10^7$-fold difference in CFU (vs. DK801) between WT and Δ*traA* strains (*Figure 7A*). These results indicate that the ability to deliver SitA3 is a

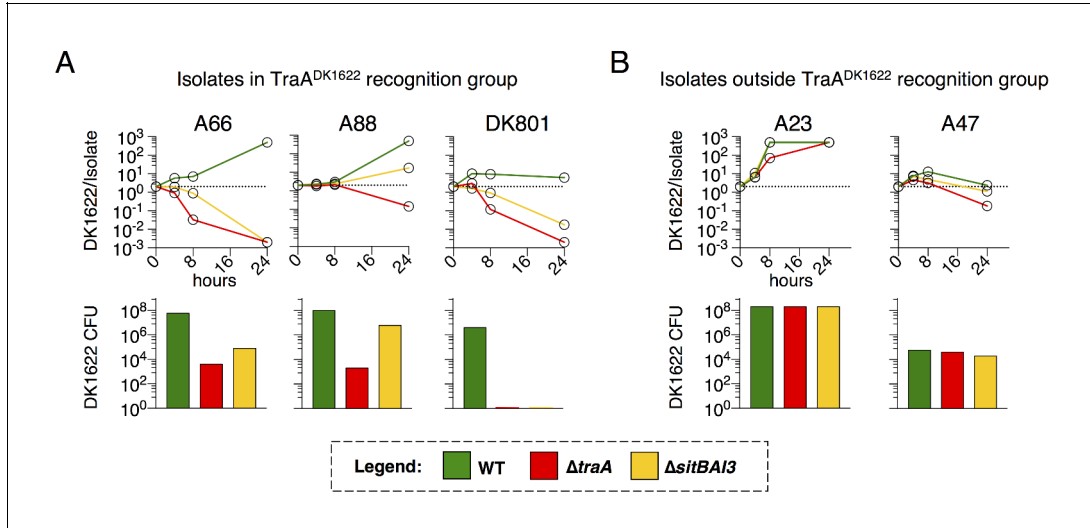

**Figure 7.** TraA and SitA are dominant determinants of competitive outcome within TraA recognition groups. (**A**) Line graphs represent strain ratio over time when the three indicated, DK1622-derived strains, which were fluorescently labelled, were competed with wild isolates (A66, A88, DK801). These isolates belong to the same TraA recognition group as DK1622. Histograms indicate viable cells (CFU) of the DK1622-derived strains (Tc$^r$) after the 24 hr competition. (**B**) Identical experiments as in A, except the lab strains were competed with wild isolates that belong to different TraA recognition groups.

dominant determinant of competitive fitness under these conditions. Not surprisingly, the finding that environmental isolates outcompeted and killed Δ*traA* strains (*Figure 7A*), confirms the existence of OME-independent killing mechanisms at play. Interestingly, these experiments revealed that Δ*traA* cells had less competitive fitness than Δ*sitBAI3,* indicating a competitive fitness defect in Δ*traA* cells beyond just the inability to deploy SitA3. As a control, we competed the DK1622 genotypes against isolates A23 and A47 (from Tübingen [*Vos and Velicer, 2006*]), which are outside the DK1622 TraA recognition group (*Pathak et al., 2013*). Δ*traA* and Δ*sitBAI3* genotypes resulted in similar competitive outcomes as WT as would be expected when inter-strain OME does not occur and SitA3 cannot be deployed (*Figure 7B*). These results show that SitA contributes significantly to fitness during competition with TraA-compatible non-kin genotypes, and likely plays a key role in competition and survival in nature.

## Discussion

### SitA promotes kin discrimination of OME partner cells

Here, we describe a novel family of proteins that carry polymorphic toxin domains and are delivered between myxobacteria by OME. SitA proteins are similar to other polymorphic toxins in that they carry diverse C-terminal domains, are neutralized by cognate immunity proteins, and are delivered in a cell contact-dependent manner. However, they are unique in their N-terminal domains, and in that they are lipoproteins transferred with other cargo during OME. To our knowledge, this is the first example of a polymorphic toxin system in which the toxin itself is a lipoprotein. Unlike CDI, T4SS, T6SS and T7SS toxins, there appears to be no requirement for a specialized apparatus to export the toxins. Instead, toxins are exchanged bi-directionally and simultaneously during OME. Therefore, SitA delivery likely only requires OM localization of the toxin and compatible TraA receptors. This discovery highlights the diversity of mechanisms used by bacteria to deliver polymorphic toxins.

Importantly, the SitA toxin family constitutes a second identity constraint upon OME with partner cells. For two cells to engage in OME, not only must they present compatible TraA receptors, but they must also contain immunity proteins to each other's toxins. In this 'recognize and verify' system (*Wall, 2016*), if the latter constraint is not met, then the recipient of the toxin is poisoned. TraA homotypic interactions alone are considered kind or greenbeard recognition, in which social interactions are based on a single gene locus. The finding that myxobacteria verify relatedness with *sitAI* confirms the notion that myxobacteria apply a *bona fide* kin discrimination mechanism during OME by requiring identity verification at multiple polymorphic loci. Interestingly, this system allows TraA interactions to promote contrasting behaviors – cooperative or antagonistic – depending on relatedness. Either outcome makes OME potentially beneficial regardless of the partner by conferring the ability to both share goods with clonemates and poison non-kin.

SitA delivery range is restricted to within a single TraA recognition group. Considering *traA* allele diversity is high (*Pathak et al., 2013*), this significantly limits the use of SitA to related but nonself individuals. This suggests that one of the primary functions of SitA is the discrimination of exchange partners, consistent with the notion that sharing large amounts of goods with non-kin is costly. Within TraA recognition groups, OM material is a shared good; a resource to be guarded from exploitation by OME compatible, yet nonself populations. Myxobacteria achieve this safeguard by inextricably linking the delivery of these goods with the delivery of SitA toxins. Another example is the *Burkholderia thailandensis* CDI system, which couples a communication signal with polymorphic toxin delivery during biofilm formation (*Anderson et al., 2014*). Similarly, the CDI system of *E. coli* mediates both antagonism and cooperative intercellular adhesion to related cells (*Ruhe et al., 2013*, *2015*). In *Proteus mirabilis*, IdsD/IdsE interactions communicate identity and may promote cooperative behaviors (*Cardarelli et al., 2015*). This intercellular communication is also coupled to toxic T6SS effector delivery (*Wenren et al., 2013*). In these examples, organisms link goods, signals, and/ or cooperative behaviors to polymorphic toxin delivery, which ensures that potential cooperators are related.

### SitA diversity and myxobacterial ecology

Differential acquisition of antagonistic systems can affect cooperation compatibility between originally identical genotypes. We demonstrated that when two otherwise isogenic colonies express

different SitA toxins, they are no longer able to merge swarms. The acquisition of a single *sitAI* operon would therefore alter strain identity and population structure between previously clonal cells. However, the impact of SitA on strain identity and population structure in natural soil habitats depends on two factors: (1) strains that belong to the same *traA* recognition group must exist in proximity, and (2) *sitA* loci must be sufficiently diverse to ensure different toxin/immunity types are represented at fine geographic scales. Velicer and colleagues have examined the compatibility of natural *M. xanthus* isolates obtained from a centimeter-scale plot of soil (*Vos and Velicer, 2009*; *Wielgoss et al., 2016*). Colony merger assays between these geographically proximal strains reveal compatibilities among the most closely related isolates, but also strong incompatibilities between strains that differ by only several dozens of mutations outside of the Mx-alpha region (*Wielgoss et al., 2016*). Indeed, these incompatibilities correlate with gene variation at hyper-variable Mx-alpha loci, where *sitBAI* genes commonly reside. Our analysis of their published sequences reveals 69 total and 15 unique *sitA* alleles distributed over 22 isolates (see Materials and methods for search criteria). Between these strains there are two *traA* alleles that are known to be incompatible (*Pathak et al., 2013*). Within each TraA recognition group we have observed an apparently high degree of correlation between the published colony merger compatibility of the strains (*Wielgoss et al., 2016*) and the *sitA* genes they possess. This suggests that *sitBAI* polymorphisms contribute to swarm incompatibility and inter-strain competition among natural soil isolates. We are currently investigating this possibility. Moreover, because Mx-alpha produces defective phage particles that promote specialized transduction (*Starich and Zissler, 1989*; *Starich et al., 1985*), these elements are apparent hotspots for HGT, perhaps explaining the high degree of Mx-alpha variation discovered between otherwise related isolates (*Wielgoss et al., 2016*). These variations likely contribute to the emergence of new compatibility types which underlie complex population structures, and explain the observation of rapidly evolving social antagonisms in *M. xanthus* (*Velicer and Vos, 2009*). Thus, TraA and SitA may act as powerful evolutionary drivers of myxobacteria diversification.

sitAI genes are often associated with prophage-like elements or other mobile elements and thus are likely acquired by HGT. By conferring a fitness advantage to their host, they may play an important role in transmission and retention of mobile DNA. For example, a HGT event into one cell in a population endowing it with a novel *sitBAI* operon allows that cell to infect its clonemates with toxins, thus ensuring the propagation of that element within the population. Similarly, the loss of the element would be lethal because the surrounding cells harbor this toxin-immunity pair would kill susceptible cells. Importantly, this model explains why many lab strains have stably maintained three large tandem repeats of Mx-alpha, which is expected to be genetically unstable (*Starich and Zissler, 1989*; *Roth et al., 1996*). In cases in which strains have spontaneously lost Mx-alpha units (*Dey et al., 2016*), those events likely occurred during propagation in liquid media, where OME cannot occur. Mx-alpha has the attributes of a selfish or addictive element that exploits the social nature of myxobacteria and OME. However, the origin of *sitBAI* loci is unclear because these genes also reside outside of selfish elements in myxobacterial genomes.

## An infectious polymorphic toxin system

SitA toxins are uniquely powerful determinants of identity, likely because they are transmitted as infectious agents between recipient cells. The infectious model is consistent with the observations that individual *M. xanthus* cells typically make contact with multiple cells simultaneously within a swarm, OME is constitutively active, and that prodigious amounts of material are transferred during OME (*Nudleman et al., 2005*). Furthermore, SitA entry into the cytoplasm occurs by a secondary and uncoupled pathway to OME (*Dey et al., 2016*; *Dey and Wall, 2014*). Thus, it is possible that SitA lingers in the OM of the primary target long enough to allow transfer to secondary target cells through subsequent OME events (*Figure 8A*). Perhaps a cellular protein is required for SitA cell entry (*Dey et al., 2016*; *Dey and Wall, 2014*), but this protein is outnumbered by SitA molecules in the OM, making cytoplasmic entry a rate-limiting step. Based on these inferences, we propose two non-exclusive models that explain serial transfer: (1) Following transfer of SitA to a primary infected cell, OME with a secondary cell occurs before the full complement of SitA enters the cytoplasm of the initial recipient (*Figure 8B*); or (2) that three or more cells are engaged in OME simultaneously (*Figure 8C*). Our results suggest that SitB functions to promote serial transfer. Given that Δ*sitB* inhibitors have defects in direct transfer, serial transfer could be blocked simply by a decrease in number of SitA molecules delivered or a decrease in rate of delivery. Alternatively, SitB could stabilize SitA in

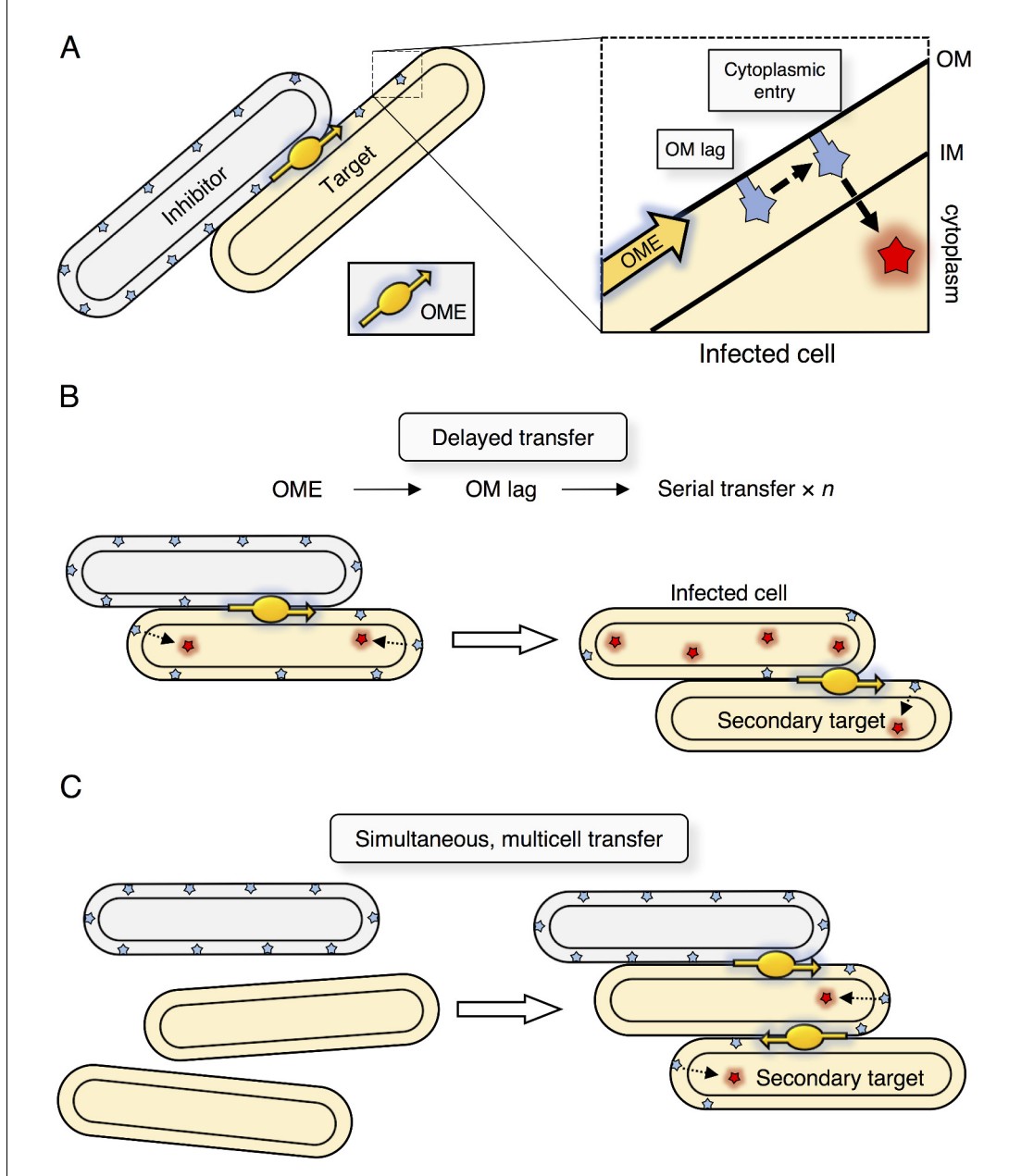

**Figure 8.** Model for serial transfer of SitA. (**A**) SitA is delivered cell-to-cell by OME. After OME, SitA may enter the cytoplasm or linger (lag) in the OM. (**B**) In the delayed entry model, the infected cell can undergo OME of SitA to another naïve cell before all SitA molecules enter the cytoplasm and before cell death. *n* = number of target cells poisoned by infected cell. (**C**) Alternative, but non-exclusive model in which OME and SitA transfer occurs between three or more cells simultaneously. Here SitA is delivered to a tertiary cell via an intermediary cell.

the OM of the inhibitor and/or target cell, thereby increasing OM dwell times to promote serial transfer. The finding that SitB is an accessory protein is consistent with bioinformatics analysis in which many of the *sitA* genes that reside outside of the *M. xanthus* species are not linked with a *sitB* gene. In the case of *M. xanthus* isolates (*Wielgoss et al., 2016*), we found only 3 of 69 *sitA* genes were not associated with *sitB*. The mechanism of serial transfer, the function of SitB, and the ability of SitA to traverse the cell envelope are topics for future study. With respect to the latter point, in prior work we found that a mutation that disrupts the inner membrane protein, OmrA, renders target cells resistant to SitA1 (*Dey et al., 2016*; *Dey and Wall, 2014*). Thus, as proposed for CDI

toxins, one possibility is that SitA toxins exploit inner-membrane proteins to gain access to the cytoplasm (*Willett et al., 2015*).

Remarkably, SitA appears to determine the competitive outcome of two-strain co-cultures between DK1622 and wild-isolates, despite the T6SS and a host of other antagonistic machinery at play (*Konovalova et al., 2010*; *Smith and Dworkin, 1994*). We hypothesize that the rapid spread of SitA by serial transfer may disrupt the antagonistic capabilities of competitors. When deployed between converging swarms, serial transfer provides a mechanism to inhibit cells behind the front-lines. Thus, SitA mediated antagonism results in the formation of distinct territorial boundaries (see *Figure 4*), which in turn minimize OME, resource sharing, and social interactions between non-clonal swarms.

### A solution to Crozier's Paradox

The selective pressure from SitA antagonism within a TraA recognition group may help drive the generation and fixation of *traA* polymorphisms that determine recognition specificity. For instance, we recently showed that simply substituting a single amino acid residue in TraA can alter recognition specificity while retaining OME function (*Cao and Wall, 2017*). More broadly, it is a puzzle how organisms select and maintain genetic variation in social genes such as *traA*. Diversification of beneficial greenbeard genes is theorized to be selected against because social groups with more common alleles receive benefit more often than those with less common alleles. This pressure to possess the allele that gains the most benefit is thought to ultimately erode allele diversity that originally allowed discrimination. This problem is known as Crozier's paradox (*Strassmann et al., 2011*; *Crozier, 1986*). Our results provide one solution to this paradox in that variation at a second locus (*sitA*, a 'harming greenbeard' [*Gardner and West, 2010*]) exerts selective pressure to diversify TraA, a helping greenbeard (*Wall, 2016*). For example, if one strain is killed by another via SitA, a TraA mutation that alters specificity within the losing population would be immune and retain OME function, and would thus be selected for. We suggest that other greenbeard systems could involve a similar balance between antagonism and cooperation that promotes maintenance of diversity for beneficial greenbeard genes.

### Concluding remarks

Our results provide a description of a novel polymorphic toxin system that helps direct cellular goods to clonemates to promote multicellular cooperation. Polymorphic toxin systems are widely prevalent in bacteria and their role in population structure, ecology and evolution of microbes is only beginning to be understood. Importantly, this study highlights the diversity of delivery mechanisms for these domains and how they have adapted to the lifestyle of their host genomes.

## Materials and methods

### Growth conditions

All strains are listed in *Supplementary file 2A*. *M. xanthus* was routinely grown in CTT medium [1% casitone; 10 mM Tris·HCl (pH 7.6); 8 mM MgSO$_4$; 1 mM KH$_2$PO$_4$] in the dark at 33°C. *E. coli* TOP10 and MG1655 were grown in LB media at 37°C. As needed for selection or induction, media were supplemented with kanamycin (50 µg/mL), oxytetracycline (10 µg/mL), ampicillin (100 µg/mL), streptomycin (50 µg/mL), arabinose (0.2%), or IPTG (1 mM). TPM buffer (CTT without casitone) or PBS was used to wash cells. CTT or LB agar was used as a solid growth medium for routine strain maintenance. For all assays, strains were grown to logarithmic growth phase, washed, and re-suspended to the appropriate density.

### Cloning and strain construction

All plasmids and primers are listed in *Supplementary file 2B,C*. Plasmids were constructed and maintained in *E. coli* and subsequently electroporated into *M. xanthus*. In the case of cloning IPTG-dependent *sitA-CTDs*, plasmids were maintained in XL1-Blue, which overexpresses LacI and reduces clone toxicity. Insertion mutations were created by amplifying an approximately 500 bp fragment of the gene of interest by PCR and cloning the fragment into the pCR-TOPO XL or pCR-TOPO 2.1 vectors (Invitrogen, Carlsbad, CA). For gene expression in *M. xanthus*, we cloned the appropriate gene

(s) into pMR3487 using XbaI and NdeI restriction sites and T4 DNA ligase. If the gene(s) of interest contained these restriction sites, we used Gibson Assembly (*Gibson et al., 2009*) (New England Biolabs, Ipswich, MA) for plasmid construction. pMR3487 recombines at a specific site in the *M. xanthus* chromosome and expression is induced with IPTG (*Iniesta et al., 2012*). Traditional restriction endonuclease cloning was used to create pBAD30, pCH450 and pKSAT derived plasmids in which the *sitA-CTD* fragments had an ATG start codon engineered into the insert. Expression of *sitI* genes in pKSAT is constitutive, driven by the $Km^r$ promoter. pCV10 for GFP expression was created by ligating tandem rRNA promoters, used for expressing *lacI* from the pMR3487 plasmid, with EGFP and the pSWU19 plasmid backbone. This plasmid recombines into the *M. xanthus* chromosome at the Mx8 phage attachment site. The deletion of *sitBAI3* was constructed by cloning in-frame regions flanking and partially overlapping the start and stop codons of *sitB3* and *sitI3,* respectively, into pBJ114 using Gibson Assembly. After recombination in *M. xanthus*, mutants were grown in CTT for 24 hr and plated on CTT containing 2% galactose to select for spontaneous loss of the *galK* marker. Deletion mutants were distinguished from WT by PCR with primers that flanked the deletion site.

## *M. xanthus* competition experiments

Competition experiments, unless otherwise noted, were done using 1:1 strain mixtures of $3 \times 10^8$ cells per mL spotted (20 μL) on agar plates containing 0.5× CTT with 2 mM $CaCl_2$ and 1 mM IPTG (competition media). Culture spots were harvested at 24 hr and observed on glass slides by microscopy to quantify strain ratios based on fluorescent labels. Typically, between 200 and 800 cells were counted. Competitive index in all assays was quantified by calculating the change in ratio of target to toxin-producing inhibitor cells over 24 hr. For example, if the 0 hr ratio was 1 to 1 and the 24 hr ratio was 1 to 100, the competitive index was. 01, indicating that the target strain was outcompeted. Swarm inhibition experiments were done identically but cells were not collected and instead were imaged after 72 hr. To determine the potency of killing, competition assays were conducted where the target cell volume and density were held constant (50 μL, $3 \times 10^8$ cells per mL) while the number of toxin producing cells were titrated 1 to 10 for each sample. Cells were harvested at 48 hr, serially diluted and plated on CTT containing Km to enumerate viable target cells.

For the SitA1 serial transfer assay, competition was done at 1:5 or 1:5:5 mixtures of inhibitors to target(s) using a culture density of $3 \times 10^9$ cells per mL. For competition with environmental isolates, liquid cultures of the DK1622-derived strain and the environmental isolate were adjusted to $3 \times 10^9$ cells per mL liquid culture, mixed (100 μL DK1622 to 50 μL isolate), and spread onto competition media to ensure there were enough non-lysed cells to count by fluorescent microscopy. At the indicated time points, 2 mL of TPM was added to the agar plate, agitated with a plate spreader, and collected by pipette. Cells were centrifuged at low speed to help prevent clumping and either cell ratio was quantified by microscopy (as above) or the cells were resuspended in 1 mL TPM for CFU determination. Clumping was not an issue due to a non-isogenic mix of strains and massive cell lysis interfering with cell-cell adhesion. CFU of DK1622-derived strains were enumerated by 1 to 10 serial dilution and plating on CTT with oxytetracycline. All figures that contain error bars indicate the experiments were done in triplicate on different days. All statistical tests comparing two results are unpaired, two-tailed *t*-tests.

## *E. coli* experiments

To test for growth inhibition by toxin expression, overnight cultures of each strain grown in LB with appropriate antibiotics were adjusted to $OD_{600} = 1.0$ and back diluted 1 to 10 into fresh media containing 0.2% arabinose (to induce expression from pBAD), ampicillin 100 μg/mL and streptomycin 50 μg/mL. Cultures were then incubated in a shaker at 37°C for 5 hr. 1 to 10 serial dilutions of each culture were plated on LB with 0.2% arabinose and 100 μg/mL ampicillin and imaged after overnight growth at 37°C.

For DAPI staining, cells were grown as described above. At 0 hr and 6 hr post arabinose induction cells were adjusted to $OD_{600} = 0.4$ and 1 mL was collected by centrifugation. Cells were fixed in freshly prepared 4% (vol/vol) formaldehyde in PBS for 15 min, rotating at room temperature (RT). The reaction was quenched by addition of an equal volume of 250 mM glycine (pH 7.5) in PBS. Cells were collected and washed 3x and resuspended to 100 μL. Cells were then spotted on poly-L-lysine coated slides and incubated for 10 min. Excess liquid was removed and cells were rinsed with water.

5 µL of fluorogel-II with DAPI (Electron Microscopy Sciences, Hatfield, PA) was placed on cells, a coverslip was applied and cells were imaged at 100× magnification with a Nikon E800 microscope coupled to a digital imaging system (*Wei et al., 2011*).

To assess tRNA processing, total RNA was isolated from *E. coli* clones using guanidine isothiocyanate-phenol extraction as described (*Garza-Sánchez et al., 2006*). Here, cultures were grown to mid-log phase and then diluted to $OD_{600}$ = 0.05 and grown for 30 min before 0.4% arabinose was added to induce toxin expression. tRNAs were analyzed by Northern blot hybridization using the following 5′-radiolabeled oligonucleotide probes: $tRNA_{UGC}^{Ala}$ (5′ - TCC TGC GTG CAA AGC AG), $tRNA_{ICG}^{Arg}$ (5′ - CCT CCG ACC GCT CGG TTC G), $tRNA_{CGA}^{Ser}$ (5′ - GTA GAG TTG CCC CTA CTC CGG), and $tRNA_{GCA}^{Cys}$ (5′ - GGA CTA GAC GGA TTT GCA A).

## Swarm merging assay

These assays used a modified competition media (replacing 1.5% agar with 1.0% agarose for imaging and addition of 5 µg/mL oxytetracycline). A multichannel pipette was used to simultaneously pipette competing strain suspensions ($1.5 \times 10^9$ cells per mL) until culture spots were nearly touching. Aliquots were air dried and plates were then incubated in a humid chamber at 33°C for 3 days. Spots were viewed on an Olympus SZX10 stereomicroscope coupled to a digital imaging system.

## Sequence analysis

To discover SitA homologs, we performed BLAST (*Altschul et al., 1990*) analysis against the IMG database (*Markowitz et al., 2012*), using the conserved N-terminal sequence (first 507 amino acids of SitA1 and the first 441 amino acids of SitA3) as queries. HMMER (*Finn et al., 2011*), HHPred (*Söding et al., 2005*) and I-TASSER (*Zhang, 2008*) analysis were performed as described above using default parameters.

To detect any correlation between colony merger and *sitA* genes from the referenced study (*Wielgoss et al., 2016*), we first identified SitA homologs by performing local BLAST analysis of the sequences published at http://www.odose.nl/u/michiel/h/22-myxo-genomes-w-annotation (*Wielgoss et al., 2016*) using SitB1 as the query. Any gene located downstream of a SitB homolog that contained the described features of SitA was considered a SitA homolog. We also used SitA homolog sequences as queries to find any *sitA* genes without an accompanying *sitB*. Redundant sequences were removed using NCBI FASTA Tools Unique Sequences webpage. The sequences were then clustered according to >96% pairwise amino acid identity and each cluster was considered a unique polymorphic toxin group. The two closest toxin groups were 90% identical. This analysis was done blind with respect to the published colony merger compatibility types (*Wielgoss et al., 2016*).

## Fluorescent transfer experiments

Log-phase reporter and target cell liquid cultures were adjusted to $1.5^{-3} \times 10^9$ cells per mL, mixed at the indicated ratios, and plated on ½ CTT agar with 2 mM $CaCl_2$, +/- IPTG, as needed. Strains were incubated at 33°C for the indicated time-period, collected, and visualized by fluorescent microscopy as described (*Wei et al., 2011*).

## Immunofluorescence and blotting

Log phase cultures were resuspended to $6 \times 10^8$ cells per mL in reduced osmolarity PBS solution ("mPBS" = 7.2 mM NaCl, 5.4 mM KCl, 10 mM $Na_2HPO_4$, 1.8 mM $KH_2PO_4$). 100 µL of 10% paraformaldehyde solution (in mPBS) and 1 µL of 5% glutaraldehyde solution (in $H_2O$) were added to 400 µL of cell suspension. Each mixture was spotted on a poly-lysine coated slide. Fixation proceeded for 30 min at RT. After rinsing with mPBS, cells were permeabilized with 0.025% Triton X-100 for 10 min and washed. Cells were blocked for 1 hr at RT with 4% BSA in mPBS, then probed with a 1:1000 final concentration of anti-FLAG antibody (in 4% BSA, Sigma, St. Louis, MO) for 1 hr at RT and subsequently washed 2× for 5 min and 1× for 10 min with mPBS. Cells were then probed with a 1:2000 final concentration of secondary antibody (in 4% BSA, Alexa Fluor 488-conjugated donkey anti-rabbit IgG; Jackson ImmunoResearch, Westgrove, PA) for 30 min at RT and subsequently washed as before. SlowFade Gold antifade reagent (Invitrogen) was added to the slide and the cells were visualized with a 100× objective lens. Western blot was performed according to standard protocols,

using anti-FLAG antibody described above, and horseradish peroxidase-conjugated goat anti-rabbit secondary antibody (Thermo Scientific, Waltham, MA).

## Expression of SitA-CTDs in *M. xanthus*

Each SitA-CTD was expressed from an IPTG-inducible promoter. For liquid growth-inhibition, cells were grown to log-phase and diluted to $5 \times 10^7$ cells per mL in two separate flasks containing fresh media (CTT, 2.5 µg/mL oxytetracycline). To one of two flasks, IPTG was added to 1.0 mM. Cells were grown in the dark with shaking at room temperature. Culture growth was monitored at the indicated time points by measuring turbidity using a Klett meter. For DAPI staining, cells were grown as above for 30 hr, washed and resuspended in mPBS to a concentration of $1.5 \times 10^9$ cells per mL. 1 µL of a 50 µg/mL DAPI solution (Life Technologies, Carlsbad, CA) was added for each mL of culture. Cells were incubated for 20 min in the dark with rotation, washed, concentrated, and visualized by fluorescent microscopy.

## Acknowledgements

We would like to thank Dr. Greg Velicer and Dr. Sébastien Wielgoss for constructive comments.

## Additional information

### Funding

| Funder | Grant reference number | Author |
|---|---|---|
| National Institute of General Medical Sciences | GM101449 | Daniel Wall |
| National Institute of General Medical Sciences | Wyoming INBRE | Daniel Wall |

The funders had no role in study design, data collection and interpretation, or the decision to submit the work for publication.

### Author contributions

CNV, Conceptualization, Data curation, Formal analysis, Validation, Investigation, Methodology, Writing—original draft, Writing—review and editing; PC, Conceptualization, Data curation, Formal analysis, Methodology; AC, HF, Data curation, Validation, Methodology, and Data generation; CSH, Conceptualization, Formal analysis, Supervision, Writing—review and editing; DW, Conceptualization, Resources, Data curation, Formal analysis, Supervision, Funding acquisition, Validation, Writing—original draft, Project administration, Writing—review and editing

### Author ORCIDs

Christopher N Vassallo, http://orcid.org/0000-0001-6992-5894
Pengbo Cao, http://orcid.org/0000-0001-6599-8121
Christopher S Hayes, http://orcid.org/0000-0002-2216-6445
Daniel Wall, http://orcid.org/0000-0002-0273-1371

## Additional files

### Supplementary files

• Supplementary file 1. Homologs of SitA toxins.

• Supplementary file 2. Strains, plasmids, and primers used in this study.

• Supplementary file 3. Alignment of representative SitA toxins and their predicted functional annotations.

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
