## [Decision Letter]

[Editors’ note: a previous version of this study was rejected after peer review, but the authors submitted for reconsideration. The first decision letter after peer review is shown below.]

Thank you for submitting your work entitled "Infectious polymorphic toxins delivered by outer membrane exchange discriminate kin in myxobacteria" for consideration by *eLife*. Your article has been reviewed by three peer reviewers, one of whom is a member of our Board of Reviewing Editors and the evaluation has been overseen by a Senior Editor. The following individuals involved in review of your submission have agreed to reveal their identity: John Whitney (Reviewer #2); Karine A. Gibbs (Reviewer #3).

Our decision has been reached after consultation between the reviewers. Based on these discussions and the individual reviews below, we regret to inform you that your work will not be considered further for publication in *eLife* at this time. However, the reviewers were all very enthusiastic about the topic and intrigued by the potential of serial transmission of an outer membrane toxin. The problem was that there were significant concerns that would have to be addressed experimentally, but could take longer than the usual two months offered for revised papers. If you are able to fully address each of the major concerns, we would welcome a resubmission. The full concerns of the reviewers are spelled out in the individual reviews below. To summarize briefly, four of the most significant concerns were a need to (i) demonstrate that the toxin domains of SitA1, SitA3, and SitA2 are toxic when expressed in *Myxococcus*, not just *E. coli*, (ii) better define the role of SitB, (iii) demonstrate that it is indeed the SitA-CTDs that are transferred, not TraA, and (iv) show directly somehow that SitA-CTDs are serially transferred in an infectious-like process.

Reviewer #1:

This paper presents an interesting initial study of SitA toxins in *Myxococcus xanthus*. The authors present data indicating that these SitA toxins may contribute to kin recognition. They further posit that these toxins are serially transferred from cell to cell like an infectious agent. This latter point is potentially quite exciting, but the data to support it are incomplete. Overall, the paper was relatively straightforward and well done, but the deficiencies in fully demonstrating that SitA toxins are serially transferred are a current, major limitation.

Major Comments:

Figure 3: It's useful to see what happens when the SitA1 and SitA3 toxin domains are expressed in *E. coli*, but I think the authors must show what happens when they (and SitA2's toxin domain) are expressed in *Myxococcus* in terms of morphology, DAPI staining, etc.

The role of SitB was unclear to me. The authors should do experiments, like those in Figure 2, in which they test strains producing only SitA and SitI, without the cognate SitB, or SitB + SitI without the cognate SitA.

Figure 5: The authors show here a 5-log drop in CFUs, but the results in Figure 2 showed only 1-2 logs drop in competitive index. This discrepancy needs to be clarified or experimentally addressed. (Also, Figure 5 should report the CFUs for the producer strain, and Figure 5 needs replicates) Perhaps the difference arises from the fact that sometimes the authors are measuring CFUs of producer and target strains and sometimes they're using fluorescence to assess competitive index?

Figure 5: The results in this figure, which are critical to the paper's primary conclusion and broad appeal about serial transfer of SitA toxins, are potentially very exciting, but incomplete. (20) The authors don't consider the possibility that the cells are exchanging Tra proteins, not SitA toxins, i.e. the intermediary strain could transfer a Tra protein to the target strain, which then enables it to be directly targeted by the producer strain. This is briefly mentioned in the Discussion, but should be addressed experimentally. (21) What happens if the intermediary carries the immunity protein, but the target strain does not? And on a related note: it's surprising how efficient the transmissibility is (e.g. Figure 5) considering that the intermediary doesn't contain the immunity protein and so should be getting inhibited/killed in the process. And if non-immune cells are really such good transmitters of the toxin, then one would expect to see a wholly different pattern in the experiments in Figure 4 showing neighboring colonies, i.e. the toxin would lead to a gradient, not clean boundaries. This should be addressed. (7) There's no direct demonstration that the SitA proteins themselves are getting transferred – this needs to be addressed experimentally. In fact, I think the authors need to show (i) that SitA proteins get transferred and (ii) that they get serially transferred. It is not beyond the scope of this paper as it is so central to the most interesting element/conclusion of the manuscript.

Related to the last point above: how do the authors envision a lipoprotein getting released from the outer membrane such that it can be transferred to another cell?

Figure 6: I don't understand these data. In Figure 6 the competition with A66 leads to a 2-3 log drop in the tra- and sitBAI3- strains, and the CFU counts show these strains present at 10^2-10^3. In contrast, in the competition with A47, the CFUs are comparable to that with A66, but there's no drop in the DK1622/isolate ratio.

Reviewer #2:

In this manuscript, Vassallo et al., examine the molecular mechanism of kin discrimination via the outer membrane exchange (OME) pathway of myxobacteria. In prior work, the authors showed that the first part of this process requires receptor compatibility via the TraA cell surface receptor. In the present study, the authors identify additional genetic criteria, in the form of toxin-immunity proteins, that determine cooperative versus antagonistic outcomes of OME. In addition, the authors provide evidence that two of these toxins, SitA1 and SitA3, possess DNase and tRNase activity, respectively, providing biochemical insight into kin discrimination. Using a three-strain competition setup and appropriate TraA/toxin-immunity alleles, the authors elegantly show that these toxins can be transferred through a TraA compatible, toxin-resistant strain to intoxicate a TraA incompatible, toxin-susceptible strain. Together, these findings represent a significant advance in our understanding of kin discrimination between bacteria facilitated by OME. Below are comments for the author's consideration:

1) The toxin-immunity concept has been well-established for several pathways involved in interbacterial antagonism. However, I'm very much intrigued by the SitB family of genes, which appear to be a unique component of *Myxococcus* OME toxin cassettes. While a detailed characterization of the proteins encoded by these ORFs is beyond the scope of this study, minimally the authors should examine whether SitB1 is required for SitA1-based antagonism and test if SitB proteins are themselves toxic when expressed heterologously. The authors mention in their concluding statement that this family of proteins is predicted to adopt a β-barrel fold. The authors should include the methods used to reach this prediction, clarify if this refers to a transmembrane or a soluble β-barrel protein and speculate on what the role of these proteins might be.

2) The lack of toxicity for SitA2 in the *E. coli* viability assays and the weak phenotype observed in competition experiments could be due to low expression of the protein. Perhaps this caveat should be mentioned in the interpretation of experiments examining SitA2-based toxicity. Alternatively, might SitA2 be toxic if expressed in the periplasm?

3) Figure 1 needs a scale bar.Ο

4) How do OME transferred toxins access the cytoplasm of recipient cells? In previous work, the authors identified mutations in omrA that blocked killing, however as noted by the authors, it is unclear if this resistance to killing is due to changes in membrane properties as a result of inactivation of OmrA enzymatic function or if OmrA is a specific receptor required for toxin import. It would be interesting to examine whether killing resistant omrA variants are resistant to intoxication by just a single SitA toxin or to all three SitA toxins. If the latter result is obtained it would provide preliminary evidence suggesting that these toxins may contain translocation domains that require specific membrane properties for cell entry.

5) It is not clear how OM tethered toxins reach the inner membrane of target cells prior to their entry into the cytoplasm. One possibility could be N-terminal cleavage of the lipid anchor, which would allow for diffusion of the toxin to the inner membrane. If the authors have an available antibody, it might be informative to look for proteolytic processing of the toxin during OME by western blot analysis.

Reviewer #3:

This Vassallo et al., manuscript is a follow-up to earlier manuscripts (Dey et al., 2016, Vassallo et al., 2015, and Dey and Wall, 2014) and helps to close the chapter on one aspect of *Myxococcus* kin recognition by answering several open questions about the role of TraA compatibility and OME during within species kin selection. In this way, this manuscript as a whole is interesting and impactful, most significantly in the *Myxococcus* field.

Individual aspects are less novel, e.g., the coupling of toxic and non-toxic signal exchange in bacteria (termed here "test and verify") has been alluded to in multiple bacterial systems including Ruhe et al., 2013 and 2015, Wenren et al., 2013, and Anderson et al., 2014 (see –Discussion section). The SitA1 and A3 toxins themselves, as well as the chromosomal organization of the cassettes, has been discussed for Rhs, CDI, and T6SS. However, the infectious agent analogy is compelling, even though spread is limited by TraA alleles; could this mechanisms be applicable to other bacterial systems?

Key aspects that are currently missing include:

– The main conclusions of the manuscript, based on the Abstract, are difficult to understand in the main text. Perhaps the Introduction and Discussion could be streamlined, focusing on one or two main ideas.

– Show the data that TraA is not being transferred. (The authors allude to unpublished data supporting this –in the Discussion section). This is crucial for the idea that only the SitA proteins go cell-cell.

– I think the authors can directly test the idea of a "carrier" population that can spread the infection without becoming susceptible, which would strengthen their argument of the Sit toxins acting like infectious agents. Briefly, the authors have previously published an omr mutant (discussed –in the Discussion section) that is immune to the SitA toxins, likely due to an inability to import the toxin into the cytoplasm (Dey and Wall, 2014). The authors could, therefore, move the omr mutation into DW2428 (DK1622 missing all sitA genes and TraA+). This strain should be immune from toxins but be able to act as a carrier. First, this strain should have the same fitness as any TraA-compatible wild-type strain, further supporting that import into the cytoplasm is key for the toxin activity. More importantly, the authors could analyze mixtures of this strain with wild-type combinations (similar to those performed in Figure 5), specifically using two different sitA-containing natural isolates of one TraA compatible group and mix with the DK1622 carrier strain. The prediction is that the "carrier" strain would have the same fitness as the wild-type strains, if not higher. This would show that a carrier strain could pass toxicity from within its TraA compatible class to any other strain regardless of its own native sit locus, which would be expected from an infection. (Consider for example a conjugation strain in a tri-parental mating.) On these same lines, do the authors frequently find omr mutants naturally arising within their populations? And wouldn't a positive outcome from this assay question the idea that this OME-sharing of toxins is a selection for kin as opposed to a just being a "true" infection?

– The authors speak to a possible correlation between sitA and traA diversity (–Discussion section, S4). This assertion would be stronger with an analysis of the alleles and the differences between the degrees of variation found in those alleles. (Or remove this assertion from the manuscript.) – Determine SitA2 function or alternatively, remove from the manuscript. The lack of discussion of SitA2 is distracting and draws away from the main points. For example in Figure 1 and – in subsection “DK1622 ancestors contain three functional *sitBAI* toxin/immunity cassettes”, the SitA2 result is not fully discussed in the manuscript.

[Editors’ note: what now follows is the decision letter after the authors submitted for further consideration.]

Thank you for resubmitting your work entitled "Infectious polymorphic toxins delivered by outer membrane exchange discriminate kin in myxobacteria" for further consideration at *eLife*. Your revised article has been favorably evaluated by Gisela Storz (Senior editor) and three reviewers, one of whom is a member of our Board of Reviewing Editors.

The manuscript has been improved, and the reviewers generally agreed that the revised paper merits publication in *eLife*, provided you are willing to address a few remaining issues raised by one of the reviewers:

Reviewer #3:

In the manuscript by Vassallo et al., the authors demonstrate that lethal activity of previously unknown toxins is a crucial component of kin discrimination. These toxins, which are located on the Mx-alpha region, are transferred during the process of outer membrane exchange (OME). The process of OME requires a separate kin-specific set of proteins, TraA and TraB. The authors propose a clever and rational model: that OME of toxic elements is similar to an infection that can spread across a bacterial population, ensuring that only those with the antidote (i.e., an inhibitor of the toxin) can survive. The authors speculate that this model could explain why TraA alleles are not sufficient to predict kin and non-kin populations and why a diversity of *M. xanthus* populations can exist in very close proximities in the wild. This is a well-executed piece of research and would be of broad scientific interest.

Major suggestions:

1) The use of the competitive index for the microscopy (Figure 2 and Figure 2—figure supplement 3) and its explanation are confusing. The authors state from –subsection “*M. xanthus* competition experiments”: "… if the 0 h ratio was one to one and the 24 h ratio was 1 to 1000, the competitive index was.001" and "Typically, between 200 and 500 cells were counted." In that case, 1000 cells were never counted so a competitive index of 0.001 should not be possible, but in Figure 2, there is a data point at > 0.001 for sitI1+ versus sitBAI3+, which I presume means "0" cells were counted.

The authors' result is clear: the target strains are outcompeted greatly. With the current competitive index values, readers have to back-calculate to get the raw numbers. However, it seems that there were very few target cells visible (<10) in most of the competitions. Reporting the raw values (or the final ratios) would be equally compelling and less confusing to the reader.

2) The use of "swarm inhibition" throughout the text seems inaccurate given that the authors have already shown that the inhibition is due to death of the targeted cells. After introducing the historical name of the genes, as the authors do in the Results –, the authors then write, "Swarm inhibition was further demonstrated to be caused by cell death of the motile strain" (–subsection “SitA1 is the swarm inhibition toxin**”**). After that point, it would be more accurate to refer to the inhibition as "growth inhibition" or "cell death," which are the precise mechanisms (and would further emphasize the functional mechanisms of the Sit toxins). This is especially true for Figure 2—figure supplement 2, because these assays were performed in liquid (not in swarms).

The authors used the lack of swarming ("inhibition") as a read-out for genetic analyses to characterize which genes were needed for inhibitory activity (Figure 1). Stating this more concisely in discussing Figure 1 would be helpful for the reader.

3) The terminology "*novel* polymorphic toxin" or "features that make them *unique among* polymorphic toxin systems" (emphases added by this reviewer) seems a bit inaccurate. The authors state that there are many systems with polymorphic toxins, and it appears from the text that the SitA1, SitA2, and SitA3 proteins each contain motifs homologous to known toxin proteins. It seems that the greatest novelty is the delivery by OME and the striking serial transfer mechanism.

4) The authors mention that sit-like genes are found in regions outside the Mx-alpha region in other myxobacteria (–subsection”Polymorphic SitA toxins are conserved in myxobacteria”). A potential hypothesis for why these regions contain sit genes should be stated. Are these regions known to have high rates of horizontal gene transfer?

---

## [Author Response]

[Editors’ note: the author responses to the first round of peer review follow.]

Reviewer #1:

*This paper presents an interesting initial study of SitA toxins in Myxococcus xanthus. The authors present data indicating that these SitA toxins may contribute to kin recognition. They further posit that these toxins are serially transferred from cell to cell like an infectious agent. This latter point is potentially quite exciting, but the data to support it are incomplete. Overall, the paper was relatively straightforward and well done, but the deficiencies in fully demonstrating that SitA toxins are serially transferred are a current, major limitation.*

The revised manuscript contains additional experiments which confirm that SitA toxins are serially transferred. First, to eliminate the possibility that TraA itself is transferred we have included Figure 5—figure supplement 1 which shows that a functional TraA-mCherry fusion is not exchanged, whereas a control OM-mCherry reporter is transferred. Second, we provide new data that strains lacking SitB are competent for direct cell-cell transfer but not serial transfer of toxins, which as described in the manuscript, strongly suggests that TraA is not transferred during the serial transfer experiments (Figure 6). Third, we now directly show by fluorescent microscopy that a SitA1-mCherry fusion is serially transferred to target cells via an intermediary cell (Figure 5).

*Major Comments:*

Figure 3: It's useful to see what happens when the SitA1 and SitA3 toxin domains are expressed in E. coli, but I think the authors must show what happens when they (and SitA2's toxin domain) are expressed in Myxococcus in terms of morphology, DAPI staining, etc.

We have added experiments in which CT domains are conditionally expressed in *M. xanthus*. The cytoplasmic CTDs inhibit growth and disrupt cell morphology and show that SitA1 and SitA2 CTD expression abolishes DAPI staining in both *E. coli* and *M. xanthus* (Figure 2—figure supplement 2).

*The role of SitB was unclear to me. The authors should do experiments, like those in Figure 2, in which they test strains producing only SitA and SitI, without the cognate SitB, or SitB + SitI without the cognate SitA.*

We have included these experiments in Figure 6 in which either SitBAI or SitAI (SitB–) expressing cells are competed against a target strain as suggested. These are done in a SitB3 mutant background to eliminate the possibility of promiscuity between SitB3 and SitA1. We found that the toxin function is defective but not abolished. However, at higher ratios of target to SitAI inhibitor, toxin function is not detected (Figure 6). These findings led us to investigate whether SitB contributed to serial transfer mentioned above.

Figure 5: The authors show here a 5-log drop in CFUs, but the results in Figure 2 showed only 1-2 logs drop in competitive index. This discrepancy needs to be clarified or experimentally addressed. (Also, Figure 5 should report the CFUs for the producer strain, and Figure 5 needs replicates) Perhaps the difference arises from the fact that sometimes the authors are measuring CFUs of producer and target strains and sometimes they're using fluorescence to assess competitive index?

There are technical problems with CFU measurements in WT backgrounds due to severe cell clumping caused by type VI pili and consequently EPS production in colony biofilms. To circumvent this issue we use pilA mutants that lacks pili, EPS and S-motility. However, we prefer to use WT backgrounds (DK1622, pilA+) and this is why we have generally used the fluorescent cell assay to measure the ratio of competing strains, i.e. competitive index. Competitive index gives us a clear answer to cell death/fitness, but has a limited dynamic range because practically speaking there are finite number of cells we can count under the microscope (~1/1000 limit). We also note that fluorescent cells with irregular morphology are counted although they are likely not viable. This is why CFU yields a broader dynamic range for cell viability and why, in part, it was used in Figure 5. The text has been modified to communicate differences in what the two assays measure. Also, you may notice that we do count WT CFUs in Figure 7. This is possible because the wild isolate and lab strain do not clump in coculture, likely due to incompatible EPS and cell death occurring.

We have now included Figure 5 data in triplicate. Although it is possible to include donor (inhibitor) strain CFU in Figure 5, we do not think this is useful. In this experiment, we titrate the number of input inhibitor cells over 5-logs, which is shown in the bar graph, and consequently their numbers will vary for each experiment. In contrast, the total number of target input cells remains constant across experiments and what is important is the CFU output number of the target cells as a function of inhibitor cell input.

Figure 5: The results in this figure, which are critical to the paper's primary conclusion and broad appeal about serial transfer of SitA toxins, are potentially very exciting, but incomplete. (20) The authors don't consider the possibility that the cells are exchanging Tra proteins, not SitA toxins, i.e. the intermediary strain could transfer a Tra protein to the target strain, which then enables it to be directly targeted by the producer strain. This is briefly mentioned in the Discussion, but should be addressed experimentally. (21) What happens if the intermediary carries the immunity protein, but the target strain does not? And on a related note: it's surprising how efficient the transmissibility is (e.g. Figure 5) considering that the intermediary doesn't contain the immunity protein and so should be getting inhibited/killed in the process. And if non-immune cells are really such good transmitters of the toxin, then one would expect to see a wholly different pattern in the experiments in Figure 4 showing neighboring colonies, i.e. the toxin would lead to a gradient, not clean boundaries. This should be addressed.

(20) We have added two experiments that clearly show that TraA is not transferred, while also showing that SitA is serially transferred. Figure 5—figure supplement 1 demonstrates that a functional TraA-mCherry fusion is not transferred. Since, in these experiments the engineered TraA-mCherry fusion was simultaneously co-expressed with TraB, we re-did all of Figure 5 experiments such that TraA and TraB was similarly co-expressed. In so doing our results did not change. Further, the data in Figure 6 shows that a SitA/I only producing strain, i.e. the *sitB* gene is deleted, can kill the direct target but is unable to serially transfer (note: here ratio of inhibitor:intermediary:target is 10:1:1, allowing efficient killing by SitA/I inhibitor on direct intermediary target). If TraA were being exchanged, we would expect SitA/I producing strain to kill both the merodiploid (intermediary) and the target, since this scheme allows direct transfer of TraA receptors from the *traA* merodiploid to the inhibitor and target cells; however the target cell was not killed, indicating that TraA itself was not transferred.

(21) We have done the experiment in which the intermediate strain is immune, either by an *omrA* (a resistant determinant that we think blocks SitA entry into the cytoplasm; see Dey et al., 2016. J. Bacteriol) mutation or by expression of SitI. In each case the intermediate strain survives but the target does not. While we understand the interest in such an experiment, we think that it is not necessary for our main conclusion, and since the paper is already long/complex, we argue that this data should be omitted.

We would like to underscore the point that OME occurs relatively quickly (a few minutes upon cell-cell contact) and delivers large amounts of OM proteins. The toxin will kill the recipient cell over time (hours; it must first enter the cytoplasm by a secondary and poorly understood pathway and death, i.e. no cell movement/metabolism, by DNase/RNase activity is not necessarily quick), but will not immediately prevent OME from further transferring toxin that may remain in the OM of the intermediate strain before the cell dies (this point was touched upon in the discussion). As for the experiments in Figure 4 gradient is unlikely to be seen because these are motile swarms. Since cell movement happens more quickly than cell killing, we are doubtful that a killing gradient would be clearly visible.

(7) There's no direct demonstration that the SitA proteins themselves are getting transferred – this needs to be addressed experimentally. In fact, I think the authors need to show (i) that SitA proteins get transferred and (ii) that they get serially transferred. It is not beyond the scope of this paper as it is so central to the most interesting element/conclusion of the manuscript.

Although all evidence shows a correlation between OME and toxin transfer, we do agree it is a formal possibility that TraA interaction and/or OME stimulates another system to deliver the toxin. To address this possibility we created a SitA1 lipobox-mCherry fusion and show that it is indeed transferred like other lipoproteins from our prior work. We also provide evidence that serial transfer happens by showing that this reporter can transfer from the producer to the target via an intermediary (*traA* merodiploid) strain (Figure 5). Additionally, by immunofluorescence we now show that an epitope-tagged, full-length SitA1 localizes to the cell envelope much like the mCherry fusion (Figure 2).

Serial transfer of the native protein is demonstrated in Figure 5 by showing that SitI protects the target from toxicity. SitI is not transferred because it is localized in the cytoplasm (our previous work) and Figure 2 also shows that SitI is not transferred. We have no alternative explanation for why SitI would protect the target if it was not SitA that was serially transferred and causes cell death. Thus, multiple lines of evidence support a serial transfer model of SitA.

*Related to the last point above: how do the authors envision a lipoprotein getting released from the outer membrane such that it can be transferred to another cell?*

As discussed in our prior work, lipoproteins are transferred from cell-to-cell in a very efficient manner (Nudleman et al., 2006 Science) by a mechanism that is thought to involve OM fusion catalyzed by TraA/B (Pathak et al., 2012 PLoS Genetics). Therefore, the lipoprotein does not have to be released from the OM, it is transferred by lateral diffusion in the membrane. The text has been expanded to clarify this point and we have added a model (Figure 8) to clearly illustrate this point.

A future question to be addressed, which is beyond the scope of this study, is how SitA, or at least the CT toxin domain, is released from the OM and enters the cytoplasm.

Figure 6 don't understand these data. In Figure 6 the competition with A66 leads to a 2-3 log drop in the tra- and sitBAI3- strains, and the CFU counts show these strains present at 10^2-10^3. In contrast, in the competition with A47, the CFUs are comparable to that with A66, but there's no drop in the DK1622/isolate ratio.

One thing to keep in mind in these experiments (now Figure 7) is that the cell ratio can remain one to one while the CFU count drops. This will happen when there is equal or bidirectional killing between the two strains, which we have confirmed is happening. To address your specific example, ∆traA and ∆sit3 strains are killed rapidly without much reciprocal killing of A66. This drives the ratio down as well as the CFU. In the case of A47, mutual and equal killing drives the CFU count down, but since A47 is also being killed equally (by a TraA-independent mechanism), the ratio remains close to 1.

Reviewer #2:

*In this manuscript, Vassallo et al., examine the molecular mechanism of kin discrimination via the outer membrane exchange (OME) pathway of myxobacteria. In prior work, the authors showed that the first part of this process requires receptor compatibility via the TraA cell surface receptor. In the present study, the authors identify additional genetic criteria, in the form of toxin-immunity proteins, that determine cooperative versus antagonistic outcomes of OME. In addition, the authors provide evidence that two of these toxins, SitA1 and SitA3, possess DNase and tRNase activity, respectively, providing biochemical insight into kin discrimination. Using a three-strain competition setup and appropriate TraA/toxin-immunity alleles, the authors elegantly show that these toxins can be transferred through a TraA compatible, toxin-resistant strain to intoxicate a TraA incompatible, toxin-susceptible strain. Together, these findings represent a significant advance in our understanding of kin discrimination between bacteria facilitated by OME. Below are comments for the author's consideration:*

*1) The toxin-immunity concept has been well-established for several pathways involved in interbacterial antagonism. However, I'm very much intrigued by the SitB family of genes, which appear to be a unique component of Myxococcus OME toxin cassettes. While a detailed characterization of the proteins encoded by these ORFs is beyond the scope of this study, minimally the authors should examine whether SitB1 is required for SitA1-based antagonism and test if SitB proteins are themselves toxic when expressed heterologously. The authors mention in their concluding statement that this family of proteins is predicted to adopt a β-barrel fold. The authors should include the methods used to reach this prediction, clarify if this refers to a transmembrane or a soluble β-barrel protein and speculate on what the role of these proteins might be.*

SitB experiments are now included. See above.

We have expressed SitB in the absence of SitA and find no toxicity.

We have added to the text the Method used to predict a transmembrane β-barrel and secretion signal for SitB and speculate on SitB function.

*2) The lack of toxicity for SitA2 in the E. coli viability assays and the weak phenotype observed in competition experiments could be due to low expression of the protein. Perhaps this caveat should be mentioned in the interpretation of experiments examining SitA2-based toxicity. Alternatively, might SitA2 be toxic if expressed in the periplasm?*

We hypothesize that the toxin domain of SitA2 may just be weaker than 1 and 3, but this does not rule out an expression issue. We have now included data where we expressed the SitA2-CTD in Myxo and showed growth inhibition. We can conclude that it is functional in the cytoplasm.

*3) Figure 1 needs a scale bar.*

Figure 1 was a schematic and all of our figures that contain micrographs have scale bars.

*4) How do OME transferred toxins access the cytoplasm of recipient cells? In previous work, the authors identified mutations in omrA that blocked killing, however as noted by the authors, it is unclear if this resistance to killing is due to changes in membrane properties as a result of inactivation of OmrA enzymatic function or if OmrA is a specific receptor required for toxin import. It would be interesting to examine whether killing resistant omrA variants are resistant to intoxication by just a single SitA toxin or to all three SitA toxins. If the latter result is obtained it would provide preliminary evidence suggesting that these toxins may contain translocation domains that require specific membrane properties for cell entry.*

We have found that *omrA* mutants confer resistance to the SitA1 class of toxins (SitA1/2/1^Mf1^) but not the SitA3 class (SitA3/3^Mf1^). Although this is an interesting topic, our work on cytoplasmic entry is still incomplete. We are currently pursuing this line of investigation and consider this topic to be beyond the scope of the current manuscript.

*5) It is not clear how OM tethered toxins reach the inner membrane of target cells prior to their entry into the cytoplasm. One possibility could be N-terminal cleavage of the lipid anchor, which would allow for diffusion of the toxin to the inner membrane. If the authors have an available antibody, it might be informative to look for proteolytic processing of the toxin during OME by western blot analysis.*

We are pursuing these type of experiments, but again we consider this to be beyond the scope of the current manuscript.

*Reviewer #3:*

*This Vassallo et al. manuscript is a follow-up to earlier manuscripts (Dey et al., 2016, Vassallo et al., 2015, and Dey and Wall, 2014) and helps to close the chapter on one aspect of Myxococcus kin recognition by answering several open questions about the role of TraA compatibility and OME during within species kin selection. In this way, this manuscript as a whole is interesting and impactful, most significantly in the Myxococcus field.*

*Individual aspects are less novel, e.g., the coupling of toxic and non-toxic signal exchange in bacteria (termed here "test and verify") has been alluded to in multiple bacterial systems including Ruhe et al., 2013 and 2015, Wenren et al., 2013, and Anderson et al., 2014 (see –Discussion section). The SitA1 and A3 toxins themselves, as well as the chromosomal organization of the cassettes, has been discussed for Rhs, CDI, and T6SS. However, the infectious agent analogy is compelling, even though spread is limited by TraA alleles; could this mechanisms be applicable to other bacterial systems?*

This is the first study to directly show that toxins are transferred by OME. Although other interesting systems, such as T6SS, also transfer toxins, our system is different in that both cells participate in exchange and it is bidirectional.

We suspect that serial transfer of toxins may occur in other systems and awaits to be discovered. We note that the requirement for myxobacteria to have compatible TraA receptors was an advantage in that it allowed us to show serial transfer in our system, whereas demonstrating serial transfer in another type of system may be more challenging.

*Key aspects that are currently missing include:*

*– The main conclusions of the manuscript, based on the Abstract, are difficult to understand in the main text. Perhaps the Introduction and Discussion could be streamlined, focusing on one or two main ideas.*

We have made general improvements to the text including reorganizing the discussion and adding section headings to guide the discussion.

*– Show the data that TraA is not being transferred. (The authors allude to unpublished data supporting this –in the Discussion section). This is crucial for the idea that only the SitA proteins go cell-cell.*

We now provide the data that TraA does not transfer. Please see above.

*– I think the authors can directly test the idea of a "carrier" population that can spread the infection without becoming susceptible, which would strengthen their argument of the Sit toxins acting like infectious agents. Briefly, the authors have previously published an omr mutant (discussed –in the Discussion section) that is immune to the SitA toxins, likely due to an inability to import the toxin into the cytoplasm (Dey and Wall, 2014). The authors could, therefore, move the omr mutation into DW2428 (DK1622 missing all sitA genes and TraA+). This strain should be immune from toxins but be able to act as a carrier. First, this strain should have the same fitness as any TraA-compatible wild-type strain, further supporting that import into the cytoplasm is key for the toxin activity. More importantly, the authors could analyze mixtures of this strain with wild-type combinations (similar to those perfomed in Figure 5), specifically using two different sitA-containing natural isolates of one TraA compatible group and mix with the DK1622 carrier strain.*

While we agree that there are several interesting experiments to be done here, we think that the current data is conclusive that serial transfer is occurring (please see above). With regard to the *omrA* mutation, we have done experiments where it acts as a “carrier”. *omrA* mutants are only resistant to the SitA1 class of toxins as discussed in response to reviewer 2. A three strain mixture in which two different strains express SitA1 or SitA2 but have incompatible *traA* alleles, but a third *traA* merodiploid strain has the *omrA* mutation were done. The *omrA* mutant overtakes the population since it is resistant to both toxins and provides a conduit for the two other strains to kill each other. We chose to leave this experiment out of the manuscript because although it is interesting, it does not add any new information to our serial transfer model.

As far as we understand the final suggested experiment, if the three strains all belong to the same TraA compatibility group there is no need for a carrier since these strains can directly transfer with each other.

Further, non-SitA antagonistic mechanisms, which we know occur in *M. xanthus* isolates, would make any result difficult to interpret.

The prediction is that the "carrier" strain would have the same fitness as the wild-type strains, if not higher. This would show that a carrier strain could pass toxicity from within its TraA compatible class to any other strain regardless of its own native sit locus, which would be expected from an infection.

This is shown in Figure 5 in which the merodiploid intermediary is a “carrier” strain. Although the carrier strain can be made resistant, as explained above, we think this does not add information to our serial transfer model.

(Consider for example a conjugation strain in a tri-parental mating.) On these same lines, do the authors frequently find omr mutants naturally arising within their populations? And wouldn't a positive outcome from this assay question the idea that this OME-sharing of toxins is a selection for kin as opposed to a just being a "true" infection?

In a screen for swarm inhibition resistant mutants (Dey and Wall, 2014), we found only *traA, traB*, and *omrA* mutants. Although *omrA* mutants do arise during this selection, they have a growth defect. This decrease in fitness may not allow fixation of *omrA* mutations in natural environments.

To your second point, we think that it is an infection in the sense of selfish DNA, i.e. a Mx-alpha prophage cell can over take sibling population by transmitting unique SitA toxin. However, from the cell's perspective this system also serves as a kin discrimination mechanism. These two scenarios are not mutually exclusive.

*– The authors speak to a possible correlation between sitA and traA diversity (–Discussion section, Figure S4). This assertion would be stronger with an analysis of the alleles and the differences between the degrees of variation found in those alleles. (Or remove this assertion from the manuscript.)*

We are suggesting that attack by polymorphic SitA toxins may help select and maintain mutations in TraA that provides recognition specificity. In other words, if one strain is killed by another via SitA, a TraA mutation in the losing population that altered specificity would survive, while retaining OME function. We hypothesize this process helps to drive and maintain diversity of TraA. In this regard, we recently showed that single amino acid substitutions can change TraA recognition specificity (Cao and Wall, 2017, PNAS). We think our hypothesis is justified and have added some text to the manuscript to improve clarity. The mentioned Figure S4 has also been removed from the manuscript.

– Determine SitA2 function or alternatively, remove from the manuscript. The lack of discussion of SitA2 is distracting and draws away from the main points. For example in Figure 1 and in subsection “DK1622 ancestors contain three functional sitBAI toxin/immunity cassettes”, the SitA2 result is not fully discussed in the manuscript.

We now show that SitA2-CTD is active when conditionally expressed in *Myxococcus* and it results in a decreased DAPI signal. We think it is important to include the SitA2 data because it shows the diversity of SitA toxins. We note that in other manuscripts toxin activity is not always experimentally demonstrated, but is included to illustrate toxin diversity. For example, function of the Tse2 toxin of *P. aeruginosa* is unknown.

[Editors' note: the author responses to the re-review follow.]

Reviewer #3:

[…]

1) The use of the competitive index for the microscopy (Figure 2 and Figure 2—figure supplement 3) and its explanation are confusing. The authors state from –subsection “M. xanthus competition experiments”: "… if the 0 h ratio was 1 to 1 and the 24 h ratio was 1 to 1000, the competitive index was.001" and "Typically, between 200 and 500 cells were counted." In that case, 1000 cells were never counted so a competitive index of 0.001 should not be possible, but in Figure 2, there is a data point at > 0.001 for sitI1+ versus sitBAI3+, which I presume means "0" cells were counted.

Yes, in this case there were no target cells left in the enumerated fields for this data point, which we have now indicated in the figure legend. We corrected the typo to now read ‘***<***’. We have also changed the hypothetical example to be more in line with most data points.

*The authors' result is clear: the target strains are outcompeted greatly. With the current competitive index values, readers have to back-calculate to get the raw numbers. However, it seems that there were very few target cells visible (<10) in most of the competitions. Reporting the raw values (or the final ratios) would be equally compelling and less confusing to the reader.*

Competitive index used here is also used in related literature, e.g. T6SS, CDI, MafB, and Cdz toxins, so we prefer to retain it. Also, CI normalizes the data with respect to the starting ratio, which maintains consistency and helps avoid reader confusion when starting ratios were changed, e.g. for Figure 6.

*2) The use of "swarm inhibition" throughout the text seems inaccurate given that the authors have already shown that the inhibition is due to death of the targeted cells. After introducing the historical name of the genes, as the authors do in the Results, the authors then write, "Swarm inhibition was further demonstrated to be caused by cell death of the motile strain" (–subsection “SitA1 is the swarm inhibition toxin**”**). After that point, it would be more accurate to refer to the inhibition as "growth inhibition" or "cell death," which are the precise mechanisms (and would further emphasize the functional mechanisms of the Sit toxins). This is especially true for Figure 2—figure supplement 2, because these assays were performed in liquid (not in swarms).*

*The authors used the lack of swarming ("inhibition") as a read-out for genetic analyses to characterize which genes were needed for inhibitory activity (Figure 1). Stating this more concisely in discussing Figure 1 would be helpful for the reader.*

We have changed some terminology to “antagonism” or “cell death” instead of “swarm inhibition”. However, since the experiment in Figure 1 measures swarming, there are some places where “swarm inhibition” use is more appropriate. To assist the reader we have modified the text to better explain that swarm inhibition is a read-out for antagonism.

*3) The terminology "*novel* polymorphic toxin" or "features that make them *unique among* polymorphic toxin systems" (emphases added by this reviewer) seems a bit inaccurate. The authors state that there are many systems with polymorphic toxins, and it appears from the text that the SitA1, SitA2, and SitA3 proteins each contain motifs homologous to known toxin proteins. It seems that the greatest novelty is the delivery by OME and the striking serial transfer mechanism.*

Whereas the C-terminal domain is shared between different toxin families, the N-terminal domain of SitA, which represents ~75% of the protein sequence, is indeed novel/unique in comparison to other toxin systems or proteins in the non-redundant database. To improve clarity, we have changed the first line of the discussion to read: “Here, we describe a novel *family of proteins* that carry polymorphic toxin domains”.

*4) The authors mention that sit-like genes are found in regions outside the Mx-alpha region in other myxobacteria (–subsection”Polymorphic SitA toxins are conserved in myxobacteria”). A potential hypothesis for why these regions contain sit genes should be stated. Are these regions known to have high rates of horizontal gene transfer?*

We now state that sit-like genes found outside of apparent Mx-alpha elements are typically located near mobile genetic elements that may have high rates of HGT.